# SPARSE REPELLENCY FOR SHIELDED GENERATION IN TEXT-TO-IMAGE DIFFUSION MODELS

## ABSTRACT

The increased adoption of diffusion models in text-to-image generation has triggered concerns on their reliability. Such models are now closely scrutinized under the lens of various metrics, notably calibration, fairness, or compute efficiency. We focus in this work on two issues that arise when deploying these models: a lack of diversity when prompting images, and a tendency to recreate images from the training set. To solve both problems, we propose a method that coaxes the sampled trajectories of pretrained diffusion models to land on images that fall *outside* of a reference set. We achieve this by adding *repellency* terms to the diffusion SDE throughout the generation trajectory, which are triggered whenever the path is *expected* to land too closely to an image in the *shielded* reference set. Our method is *sparse* in the sense that these repellency terms are zero and inactive most of the time, and even more so towards the end of the generation trajectory. Our method, named **SPELL** for *sparse repellency*, can be used either with a static reference set that contains protected images, or dynamically, by updating the set at each timestep with the expected images concurrently generated within a batch. We show that adding SPELL to popular diffusion models improves their diversity while impacting their FID only marginally, and performs comparatively better than other recent training-free diversity methods. We also demonstrate how SPELL can ensure a shielded generation away from a very large set of protected images by considering all 1.2M images from ImageNet as the protected set.

| Stable Diffusion 3 | Simple Diffusion | MDTv2 |
|:---:|:---:|:---:|
| *A close-up of an apple* | *The Eiffel Tower* | *ImageNet class 145* |

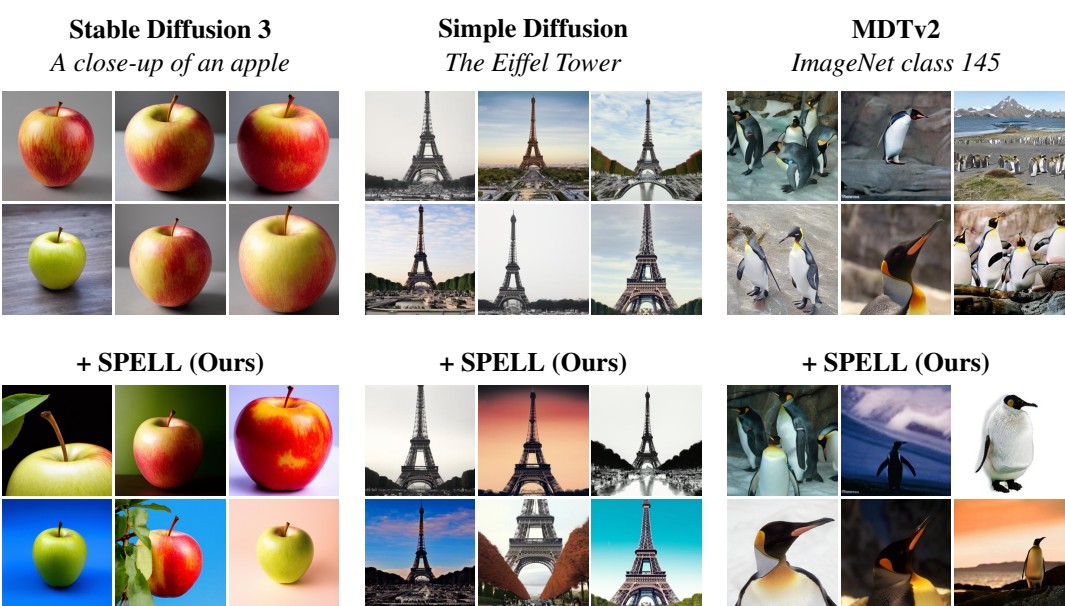

| + SPELL (Ours) | + SPELL (Ours) | + SPELL (Ours) |

Figure 1: SPELL interventions can change the diffusion trajectory of any pre-trained diffusion model, by self-avoiding other images generated within the same batch (here of size 6). SPELL achieves a higher intra-batch diversity using the same models, prompts and noise seeds.

## 1 INTRODUCTION

Diffusion models (Song et al., 2021; Ho et al., 2020) are by now widely used for engineering and scientific tasks, to generate realistic signals (Esser et al., 2024) or structured data (Jo et al., 2022; Chamberlain et al., 2021). Diffusion models build upon a strong theoretical foundation used to guide parameter tuning (Kingma & Gao, 2023) and network architectures (Rombach et al., 2022), and are widely adopted thanks to cutting-edge open-source implementations. As these models gain applicability to a wide range of problems, their deployment reveals important challenges. In the specific area of text-to-image diffusion (Nichol et al., 2022; Saharia et al., 2022), these challenges can range from an expensive compute budget (Salimans & Ho, 2022) to a lack of diversity (Ho & Salimans, 2022; Shipard et al., 2023) and/or fairness (Cho et al., 2023; Shen et al., 2024).

**Controllable Generation.** We focus on the problem of ensuring that images obtained from a model are sufficiently different from a reference set. This covers two important use-cases: *(i)* the purveyor of the model wants images generated with its model to fall *outside* of a reference set of protected images; *(ii)* the end-user wants high diversity when generating multiple images with the same prompt, in which case the reference set could consist of all previously generated images, or even other images generated concurrently in a batch. While the problem of avoiding generating images in a protected training set (Carlini et al., 2023) originates naturally when deploying products, that of achieving diversity within a batch of generated images with the same prompt should not arise, in theory, if diffusion models were perfectly trained. However, as shown for instance by Sadat et al. (2024), state-of-the-art models that incorporate classifier-free guidance (Ho & Salimans, 2022, CFG) do a very good job at outputting a first picture when provided with a prompt, but will typically resort to slight variations of that same image when re-prompted multiple times. This phenomenon is illustrated in Figure 1 for three popular diffusion models, Stable Diffusion 3 (Esser et al., 2024), Simple Diffusion (Hoogeboom et al., 2023) and MDTv2 (Gao et al., 2023).

**Contributions.** We propose a guidance mechanism coined *sparse repellency* (SPELL), which repels the backward diffusion at generation time away from a reference set of images.

- The SPELL mechanism adjusts diffusion trajectories when the generation of an image is *expected* to land close to a *shielded* region (defined as balls around the latent representation of images in the reference set). These regions can be static or updated dynamically, when used to obtain diversity, by using the *expectation* of other generation trajectories within the same batch.
- SPELL interventions are sparse in that they only consider *by design* very few active shielded images (typically one) in the reference set at each time $t$, and make these changes rarely throughout diffusion trajectory. Because they are driven by geometric considerations on the *expected* final generation output at time 0, these interventions happen mostly *early* in the backward diffusion.
- We show that applying SPELL to numerous state-of-the-art open-sourced diffusion models leads to images that better reflect the diversity of the true data (see Figure 1) with a marginal or even no increase in the Fréchet inception distance (FID) between generated images and training dataset.
- SPELL is simply parameterized by $r$, the shields' radius. We show that increasing $r$ increases accordingly the output's diversity, with a better diversity-precision trade-off than other recently proposed methods (Sadat et al., 2024; Corso et al., 2024; Kynkäänniemi et al., 2024).
- We showcase that SPELL can be scaled to a static reference set of millions of images, thanks to fast nearest-neigbor search tools. We use the whole ImageNet-1k dataset as the protected reference set, and generate images that are novel, *without* ever requiring to filter and/or regenerate images.

## 2 BACKGROUND

**Diffusion Models,** also known as score-based generative models (Song et al., 2021; Song & Ermon, 2019; Ho et al., 2020), enable sampling from data distribution $p_{\text{data}}$ on support $\mathcal{X} \subset \mathbb{R}^d$, such as an image dataset, by simulating the reverse stochastic differential equation (SDE) (Haussmann & Pardoux, 1986; Anderson, 1965), initialised from some easy to sample prior $p_1 \in \mathcal{P}(\mathbb{R}^d)$:

$$\mathrm{d}\mathbf{X}_t = [f(t, \mathbf{X}_t) - g^2(t)\nabla \log p_t(\mathbf{X}_t)]\mathrm{d}t + g(t)\mathrm{d}B_t \qquad \mathbf{X}_1 \sim p_1, \qquad (1)$$

where $(B_t)_t$ denotes Brownian motion and $p_t$ is defined as the density of $\mathbf{X}_t$ from forward process:

$$\mathrm{d}\mathbf{X}_t = f(t, \mathbf{X}_t)\mathrm{d}t + g(t)\mathrm{d}B_t \qquad \mathbf{X}_0 \sim p_0 := p_{\text{data}}, \qquad (2)$$

for drift $f : [0, 1] \times \mathcal{X} \rightarrow \mathcal{X}$ and diffusion scale $g : [0, 1] \rightarrow \mathbb{R}$, where the time $t$ is increasing in equation 2 and time decreasing in equation 1. The score term $\nabla \log p_t(\mathbf{X}_t)$ is typically approximated by a neural network through denoising score matching (Vincent, 2011).

**Training.** The solution to forward diffusion in equation 2 for an affine drift is of the form $\mathbf{X}_t = \alpha_t \mathbf{X}_0 + \sigma_t \varepsilon$, where $\varepsilon \sim \mathcal{N}(\mathbf{0}, \mathbb{I})$ for some coefficients $\alpha_t \in \mathbb{R}$, $\sigma_t \in \mathbb{R}$ (Song et al., 2021; Särkkä & Solin, 2019). The intractable score term may be expressed via denoiser using Tweedie's formula (Efron, 2011; Robbins, 1956): $\nabla \log p_t(x_t) = \frac{\alpha_t \mathbb{E}[\mathbf{X}_0 | \mathbf{X}_t = x_t] - x_t}{\sigma_t^2}$. Hence rather than estimating the score directly, one may approximate $\mathbb{E}[\mathbf{X}_0 | \mathbf{X}_t = x_t]$ via regression, by minimizing:

$$\theta^\star := \arg\min_\theta \mathbb{E}_{\mathbf{X}_t, \mathbf{X}_0} \| D_\theta(t, \mathbf{X}_t, y) - \mathbf{X}_0 \|^2 \tag{3}$$

known as *mean*-prediction, for optional condition denoted $y$, then estimate the score via $\nabla \log p_t(x_t \mid y) \approx s_{\theta^\star}(t, x_t, y) := (\alpha_t D_{\theta^\star}(t, x_t, y) - x_t)/\sigma_t^2$. Notice that we do not train model parameters in this work, and will always assume that $\theta^\star$ is given by the purveyor of a model.

**Conditioning and Guidance.** Conditional generation of diffusion generative models requires access to the conditional score $\nabla \log p_t(x_t \mid y)$ for some condition $y$ such as text or label. It is typically approximated either with explicit conditioning during training of the score / denoising network using equation 3 or as post-hoc additional guidance term added to the score. Given diffusion models have lengthy training procedures, likely due to their high variance loss (Jeha et al., 2024), it is desirable to guide diffusion models with inexpensive post-training guidance (Dhariwal & Nichol, 2021; Zhang et al., 2023; Denker et al., 2024), using e.g. classifier guidance (Dhariwal & Nichol, 2021)

$$\nabla \log p_t(x_t \mid y) = \nabla \log p_t(x_t) + \gamma \nabla \log p_t(y \mid x_t) \tag{4}$$

whereby the gradient $\nabla \log p(y \mid x_t)$ of classifier $p(y \mid x_t)$ for label $y$ is added to the score, heuristically multiplied by a scalar $\gamma \geq 1$ for increased guidance strength. Another approach which circumvents training a time-indexed classifier is using the approximation $p(y \mid x_t) \approx p(y | \mathbf{X}_0 = D_{\theta^\star}(t, x_t))$, for a pretrained denoiser $D_{\theta^\star}$ alias Diffusion Posterior Sampling (Chung et al., 2023).

**Classifier-Free Guidance and Lack of Diversity in text-to-image Diffusion Models.** Classifier-free guidance (CFG) (Ho & Salimans, 2022) is the dominant conditioning mechanism in text-to-image diffusion models, sharing properties with both explicit training and guidance. Similar to classifier guidance, CFG may be used to increase guidance strength but without resorting to approximating density $p(y \mid x_t)$. Instead, the difference $\nabla \log p(y \mid x_t) = \nabla \log p(x_t \mid y) - \nabla \log p_t(x_t)$ is used as a guidance term, where each term is approximated with the same conditional network: $\nabla \log p(x_t \mid y) \approx s_\theta(t, x_t, y)$ and $\nabla \log p(x_t) \approx s_\theta(t, x_t, \emptyset)$, for null condition $\emptyset$, trained as in equation 3. Adding CFG to the unconditional score yields $\nabla \log p_t(x_t \mid y) \approx \gamma s_\theta(t, x_t, y) - (\gamma - 1) s_\theta(t, x_t, \emptyset)$. Despite its widespread popularity and good performance, CFG weighting is heuristic. It is not clear what final distribution is being generated; and practitioners observe a lack of diversity in generated samples (Somepalli et al., 2023; Chang et al., 2023; Wang et al., 2024b). In order to combat the lack of diversity from CFG, Corso et al. (2024) add to the score a repulsive interacting potential $\Phi$ evaluated on the whole batch, $(x_t^{(i)})_i$, coined particle guidance: $\nabla \log \Phi_t((x_t^{(i)})_i)$. This repulsive potential is parameterised through a kernel as $\log \Phi_t((x_t^{(i)})_i) = -\sum_{i,j} k_t(x_t^{(i)}, x_t^{(j)})$. They specifically study the Gaussian kernel, Green's function $k(x, y) \propto 1/\|x - y\|^N$ or kernels combined with a feature extractor $e_\phi$ such as DINO (Caron et al., 2021) $k_t(x, y) = k_t^e(e_\phi(x^{(i)}), e_\phi(y^{(j)}))$.

## 3 SPARSE REPELLENCY

We propose a mechanism to sample from the data distribution $p_0$ whilst satisfying the important requirement that generated samples $\mathbf{X}_0 \sim p_0$ are distanced from each element of the reference set of *repellency* images $z_i \in \mathcal{X}, k = 1, \ldots, K$. That set may be populated by real-world protected images, samples generated in earlier batches to increase diversity, images expected to be generated by other trajectories in the current batch, or a mix of all these types. More formally, we wish to sample a conditioned trajectory $\mathbf{X}_t \mid (\mathbf{X}_0 \notin S)$, where $S$ is the collection of *shields*, i.e. balls of radius $r > 0$ around repellency images, $S := (\cup_k B_k)$ with $B_k = \{x \in \mathcal{X} : \|x - z_k\|_2 \leq r\}$. A blunt mechanism to guarantee generation outside of $S$ is to generate and discard: resample multiple times both initial noise and Brownian samples, follow the diffusion trajectory and repeat until a generated image falls outside of $S$. In the context of computationally expensive diffusion models,

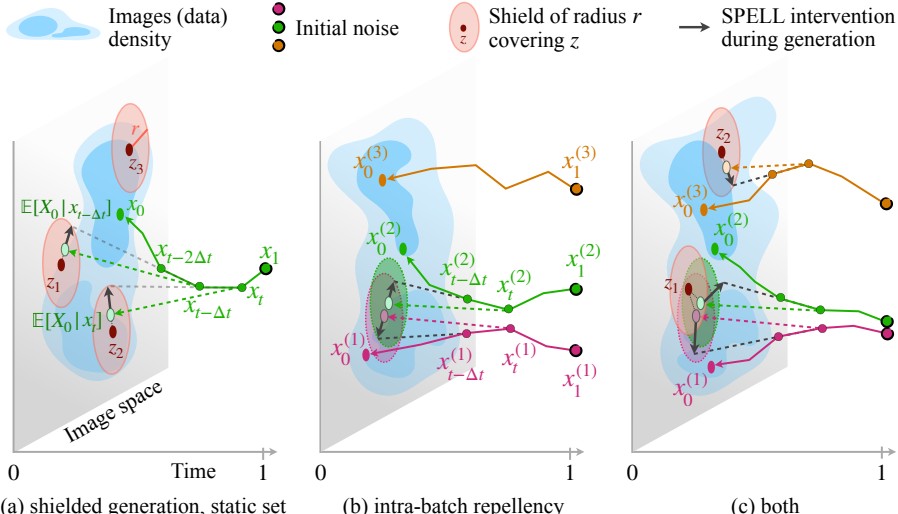

Figure 2: (a) At time $t$, by computing $\mathbb{E}[\mathbf{X}_0 \mid \mathbf{X}_t = x_t]$, we detect that the trajectory is headed (in expectation) into the shield of radius $r$ centered around $z_2$. Our sparse repellency (SPELL) term depicted as a black arrow adds a correction when generating $x_{t-\Delta t}$ to ensure that the trajectory is pushed out of the shield. This is again in the case in the next step, when starting from $x_{t-\Delta t}$. (b) In batched generation, the shields are dynamically recreated at every iteration around each trajectory's expected output. This prevents two elements in the batch, $x_t^{(1)}$ and $x_t^{(2)}$, from generating the same output. (c) Both approaches can be combined to yield a diverse set of images that won't fall into protected images and previously or concurrently generated images.

this would be wasteful and inefficient. We propose instead SPELL, a fairly simple approach that can be described from geometric principles and linked to DPS (Chung et al., 2023) and particle guidance Corso et al. (2024).

**A Geometric Interpretation of SPELL.** To ensure that generation falls outside of the shielded set $S$, our aim is to modify the diffusion trajectory at each time step, as presented in Figure 2, without having to discard any generation. To do so, we will rely at each time $t$ on the *expected* final output of the diffusion $\mathbb{E}[\mathbf{X}_0 \mid \mathbf{X}_t = x_t]$ given current state $x_t$, as approximated by the diffusion network $D_{\theta^\star}(t, x_t)$. We correct the trajectory whenever that expected vector falls within a shield. Using the notation $\hat{x}_0 = D_{\theta^\star}(t, x_t)$, we test whether for any index $k$ one has $\|\hat{x}_0 - z_k\|_2 < r$. If that is the case, the minimal modification $\delta$ that can ensure $\|\hat{x}_0 + \delta - z_k\|_2 \geq r$ is

$$\delta_k(\hat{x}_0) := \frac{(\hat{x}_0 - z_k)r}{\|\hat{x}_0 - z_k\|_2} - (\hat{x}_0 - z_k).$$

Across all $k = 1, \ldots, K$, we modify the trajectory only for those $k$ that $\hat{x}_0$ is too close to, giving

$$\Delta = \sum_{k=1}^{K} \mathbf{1}_{\hat{x}_0 \in B_k} \cdot \delta_k(\hat{x}_0) = \sum_{k=1}^{K} \text{ReLU}\left(\frac{r}{\|\hat{x}_0 - z_k\|_2} - 1\right) \cdot (\hat{x}_0 - z_k) \in \mathbb{R}^d \tag{5}$$

where the set of indices $k$ that the ReLU is non-zero for at each individual timestep is usually very small. Under the assumption that all of their shields $B_k$ are disjoint, for example when the radius $r$ is small enough, this update strictly ensures that $\hat{x}_0 + \Delta \notin S$. When shields overlap, we do not have such a guarantee. While more complicated projection operators might still yield exact updates in that case, they would involve the resolution of quadratic program. We take the view in this paper that $\Delta$ strikes a good balance between accuracy and simplicity.

Figure 2(a) visualizes the repellence mechanism away from protected images while Figure 2(b) shows how it repels from trajectories within the same batch to enhance diversity. The batch generation produces $B$ samples $x_0^{(b)}$ in parallel and its repellency mechanism uses a time-evolving set of repellency points $z_{k,t} = \mathbb{E}[\mathbf{X}_0 \mid \mathbf{X}_t = x_t^{(k)}]$ corresponding to the currently predicted end-state

of each sample in the batch. As we made no further assumptions on $z_k$, these mechanisms can be mixed as in Figure 2(c) to enable diverse generation across arbitrary numbers of batches. This makes it possible to generate large numbers of diverse images even when the VRAM for each batch is limited. Note that SPELL is a post-hoc method that does not require retraining and can be applied to any diffusion score, in RGB space or latent space, unguided or classifier-free guided. Appendix D provides pseudo code and further implementation details.

**SPELL as a DPS guidance term**. We propose to derive SPELL as a guidance mechanism that can be tightly related to Doob's $h$-transform. By Bayes' rule

$$\nabla_{x_t} \log p_t(x_t \mid x_0 \notin S) = \nabla_{x_t} \log p_t(x_t) + \nabla_{x_t} \log p_{0|t}(x_0 \notin S \mid x_t).$$

Hence, we may sample $\mathbf{X}_t \mid x_0 \notin S$ by adjusting a pretrained score function and simulating:

$$d\mathbf{X}_t = \left[ f(t, \mathbf{X}_t) - g(t)^2 (\nabla \log p_t(\mathbf{X}_t) + \nabla \log p_{0|t}(\mathbf{X}_0 \notin S \mid x_t)) \right] dt + g(t) dB_t. \tag{6}$$

The term $\log p_{0|t}(x_0 \notin S \mid x_t)$ in the score adjustment is known as Doob's $h$ transform, and provides a broadly applicable approach to conditioning and guiding diffusions. Unfortunately, Doob's $h$ transform is generally intractable. We may however appeal to diffusion posterior sampling (Chung et al., 2023) and approximate $p_{0|t}$ as a Gaussian with mean $\hat{x}_0 \approx \mathbb{E}[\mathbf{X}_0 \mid x_t]$, which is available from diffusion model pre-training, see Section 2. This approximation results in the following correction:

$$\nabla \log p_{0|t}(\mathbf{X}_0 \notin S \mid x_t) \approx \sum_{k=1}^{K} \omega(\|\hat{x}_0 - z_k\|_2, r) \cdot (\hat{x}_0 - z_k), \tag{7}$$

where $\omega(\cdot, r)$ is a weighting factor detailed in Appendix A that decreases in its first variable. This DPS approximation is similar to SPELL in that both push away trajectories in the directions $(\hat{x}_0 - z_k)$, weighted by a factor that depends on $r$ and the distance $\|\hat{x}_0 - z_k\|_2$. The difference is that DPS based on Gaussians provides a soft guidance that slowly vanishes as $\hat{x}_0$ moves away from $z_k$, and not a hard guarantee that we respect the protection radii around each $z_k$. We have struggled in preliminary experiments to set hyperparameters of such "softer" DPS schemes, because the weight factor to scale the Gaussian by ultimately depends on the magnitude of the likelihood of the shields, which is unknown, and because the Gaussian's weight never becomes exactly zero, hindering sparsity. This is why we focus our attention on the simpler and much cheaper SPELL.

**(Intra-batch) SPELL and Particle Guidance**. When using SPELL to promote diversity within the generation of a single batch (but without the more general protection against arbitary or previously generated images), SPELL can be related to the self-interacting particle guidance (PG) approach proposed by Corso et al. (2024). That approach defines an interacting energy potential $\phi_t$ at time $t$, using the locations in space of all $B$ particles within a batch at time $t$. The gradient of that potential w.r.t. each particle, $\Delta^{(i)} = \nabla_{x_t^{(i)}} \log \Phi_t(x_t^{(1)}, \ldots, x_t^{(B)})$ is then used to correct each individual trajectory to guarantee diversity. In contrast to this approach, SPELL draws insight on the expected *future* locations of points, at the end of the trajectory, i.e on the *expected denoised* images $\hat{x}_0^{(i)}$ and $\hat{x}_0^{(k)}$, where $\hat{x}_0 = D_{\theta^\star}(t, x_t)$. Indeed, the correction for each particle is explicitly given as:

$$\Delta^{(i)} := \sum_{k=1}^{K} \text{ReLU}\left( \frac{r}{\|\hat{x}_0^{(i)} - \hat{x}_0^{(k)}\|_2} - 1 \right) \cdot (\hat{x}_0^{(i)} - \hat{x}_0^{(k)}). \tag{8}$$

The $B$ correction terms $\Delta^{(i)}$ considered by SPELL cannot be seen to our knowledge as the gradients of an interacting potential. While we prove in Appendix C that $h(x) = \text{ReLU}(\frac{r}{\|x\|} - 1)x$ is a conservative field (i.e. the gradient of a potential), we find no guarantee for the more complex SPELL updates above which involve compositions of $h$ with the denoiser $D_{\theta^\star}$. Even if SPELL was a conservative field, the biggest difference between PG and SPELL is that PG defines dense interventions between all particles using soft-vanishing kernels that are never zero and thus always perturb diffusion trajectories. SPELL, conversely, intervenes sparsely and rarely, both in time and w.r.t. points in the reference set. As a result, the original diffusion process is less perturbed, notably towards the end of a trajectory, and SPELL can scale to large reference sets of millions of shields.

**Overcompensation**. While our method gives the exact weight required to land outside the shielded areas in Equations (5) and (8), we have experimented with scaling these $\Delta$ terms by an *overcompensation* multiplier $\lambda$. Intuitively, the larger that multiplier, the earlier the trajectory will be lead out of the shielded areas, with the possible downside of getting more hectic dynamics. We illustrate this addition in Figure 5 with a value $\lambda = 1.6$, which we find to work favorably across multiple models.

Table 1: SPELL improves the diversity of text-to-image and class-to-image diffusion models considerably, at only a small trade-off in terms of precision. The results are reported as mean ± std, computed over 5 independent runs with different seeds over the full dataset.

| Model | Setup | Recall ↑ | Vendi ↑ | Coverage ↑ | Precision ↑ | Density ↑ | FID ↓ | FD$_{\text{DINOv2}}$ ↓ |
|---|---|---|---|---|---|---|---|---|
| Latent Diffusion | text-to-image | $0.236 \pm 0.003$ | $2.527 \pm 0.005$ | $0.447 \pm 0.001$ | $\mathbf{0.559} \pm 0.000$ | $\mathbf{0.768} \pm 0.002$ | $\mathbf{9.501} \pm 0.024$ | $106.244 \pm 0.384$ |
| + SPELL (Ours) | text-to-image | $\mathbf{0.289} \pm 0.003$ | $\mathbf{2.695} \pm 0.002$ | $\mathbf{0.457} \pm 0.001$ | $0.551 \pm 0.001$ | $0.745 \pm 0.002$ | $9.554 \pm 0.043$ | $\mathbf{98.761} \pm 0.441$ |
| SD3-Medium | text-to-image | $0.379 \pm 0.004$ | $3.749 \pm 0.005$ | $\mathbf{0.294} \pm 0.000$ | $\mathbf{0.313} \pm 0.001$ | $\mathbf{0.345} \pm 0.001$ | $\mathbf{20.103} \pm 0.090$ | $\mathbf{230.248} \pm 0.812$ |
| + SPELL (Ours) | text-to-image | $\mathbf{0.483} \pm 0.002$ | $\mathbf{4.711} \pm 0.013$ | $0.229 \pm 0.001$ | $0.211 \pm 0.002$ | $0.213 \pm 0.002$ | $35.174 \pm 0.153$ | $482.246 \pm 0.948$ |
| Simple Diffusion | text-to-image | $0.230 \pm 0.003$ | $2.799 \pm 0.006$ | $0.355 \pm 0.002$ | $\mathbf{0.441} \pm 0.001$ | $\mathbf{0.556} \pm 0.002$ | $\mathbf{19.879} \pm 0.003$ | $\mathbf{245.138} \pm 0.586$ |
| + SPELL (Ours) | text-to-image | $\mathbf{0.248} \pm 0.002$ | $\mathbf{2.886} \pm 0.005$ | $0.355 \pm 0.001$ | $0.433 \pm 0.002$ | $0.541 \pm 0.002$ | $19.959 \pm 0.033$ | $245.748 \pm 0.562$ |
| EDMv2 | class-to-image | $0.589 \pm 0.002$ | $11.645 \pm 0.022$ | $\mathbf{0.551} \pm 0.002$ | $\mathbf{0.518} \pm 0.002$ | $\mathbf{1.404} \pm 0.005$ | $\mathbf{3.377} \pm 0.022$ | $\mathbf{68.452} \pm 0.298$ |
| + SPELL (Ours) | class-to-image | $\mathbf{0.600} \pm 0.002$ | $\mathbf{11.806} \pm 0.013$ | $0.547 \pm 0.001$ | $0.508 \pm 0.001$ | $1.364 \pm 0.005$ | $3.456 \pm 0.021$ | $68.909 \pm 0.161$ |
| SD3-Medium-Class | class-to-image | $0.143 \pm 0.002$ | $8.861 \pm 0.028$ | $\mathbf{0.202} \pm 0.002$ | $\mathbf{0.323} \pm 0.002$ | $\mathbf{0.801} \pm 0.005$ | $\mathbf{22.246} \pm 0.020$ | $\mathbf{328.032} \pm 0.571$ |
| + SPELL (Ours) | class-to-image | $\mathbf{0.206} \pm 0.002$ | $\mathbf{12.190} \pm 0.032$ | $0.146 \pm 0.001$ | $0.181 \pm 0.002$ | $0.420 \pm 0.006$ | $38.709 \pm 0.054$ | $478.286 \pm 0.553$ |
| MDTv2 | class-to-image | $0.623 \pm 0.002$ | $12.546 \pm 0.021$ | $0.505 \pm 0.001$ | $0.401 \pm 0.002$ | $1.020 \pm 0.002$ | $4.884 \pm 0.052$ | $133.175 \pm 0.721$ |
| + SPELL (Ours) | class-to-image | $\mathbf{0.634} \pm 0.002$ | $\mathbf{12.772} \pm 0.027$ | $0.505 \pm 0.001$ | $\mathbf{0.407} \pm 0.001$ | $\mathbf{1.029} \pm 0.005$ | $\mathbf{4.381} \pm 0.047$ | $\mathbf{122.125} \pm 0.291$ |

# 4 EXPERIMENTS

We now show that SPELL increases the diversity of modern text-to-image and class-to-image diffusion models (Section 4.2), with a better trade-off than other recent diversity methods (Section 4.3). We quantify the sparsity of SPELL interventions in Section 4.4 and utilize it to scale up to using all 1.2 million ImageNet-1k train images as a shielded set in Section 4.6.

## 4.1 EXPERIMENTAL SETUP

We add SPELL to both class-to-image and text-to-image diffusion models. In the class-to-image setup, we use Masked Diffusion Transformers (MDTv2) (Gao et al., 2023), EDMv2 (Karras et al., 2024), and Stable Diffusion 3 Medium (SD3) (Esser et al., 2024). Since SD3 is a text-to-image model, we follow their template *"a photo of a class_name"*. We use the pretrained model checkpoints to generate 50,000 256x256 images of ImageNet-1k classes(Deng et al., 2009) without and with SPELL and compare them to the original ImageNet-1k images. We use the validation dataset as a comparison, since we will conduct experiments that repel from the training dataset in Section 4.6, which would render comparisons to the training dataset meaningless. In our text-to-image setup, we use SD3, Latent Diffusion (Rombach et al., 2022), and RGB-space Simple Diffusion (Song et al., 2021) in resolution 256x256. For the latter two, we use the checkpoints of Gu et al. (2023). Details on hyperparameters are provided in Appendix D. We evaluate these models on CC12M (Changpinyo et al., 2021). As we target the ground truth diversity of images related to each prompt, we select a subset of captions with multiple corresponding images. This gives a one-to-many setup with 5000 captions and 4 to 128 images each (in total 41,596 images). We explain the construction of this dataset in Appendix E. To evaluate diversity, we track the recall (Kynkäänniemi et al., 2019), coverage (Naeem et al., 2020), and Vendi score (Friedman & Dieng, 2023). To evaluate image quality, we use precision (Kynkäänniemi et al., 2019) and density (Naeem et al., 2020). We track these metrics per class/prompt and average across classes/prompts. To measure whether the generated images match the true image distributions, we use the marginal FID (Heusel et al., 2017) and the marginal Fréchet Distance with DINOv2 features (FD$_{\text{DINOv2}}$, Stein et al. (2024); Oquab et al. (2024)).

## 4.2 BENCHMARK

We first examine whether adding SPELL post-hoc increases the diversity of trained diffusion models. To this end, we quantitatively compare each diffusion model to the same model run with the same random generation seeds but with SPELL. In particular, we use intra-batch repellency together with repellency from previously generated batches, to enable repellency across the up to 128 images per prompt/class. Table 1 shows that SPELL consistently increases the diversity, both in terms of recall and Vendi score, across all text-to-image and class-to-image diffusion models. This demonstrates that SPELL works independent of the model architecture and the space the models diffuse in (RGB space for Simple Diffusion, VAE space for all others). Coverage remains within -1% to +2% of its original value in all models except SD3. The difference between coverage and recall is that coverage uses a more tight neigborhood radius to determine whether an image of the original dataset is covered by the generated ones. In other words, coverage measures a form of dataset match, which

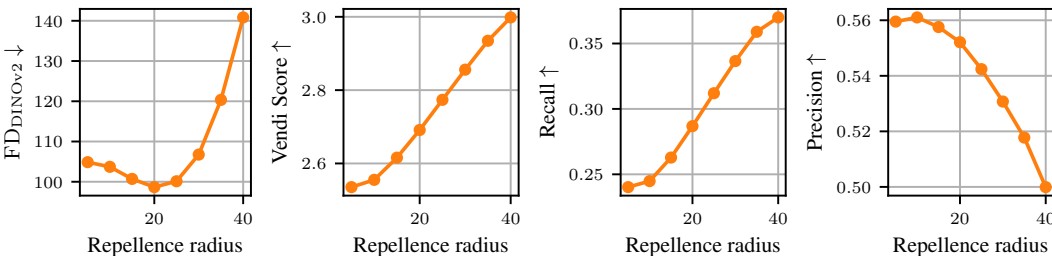

Figure 3: Effect of SPELL's hyperparameter $r$ on Latent Diffusion metrics on CC12M. A small radius ($r = 15$) improves the Vendi score, recall, and FD$_{\text{DINOv2}}$ without compromising precision. The radius can be further increased to trade-off precision for additional diversity.

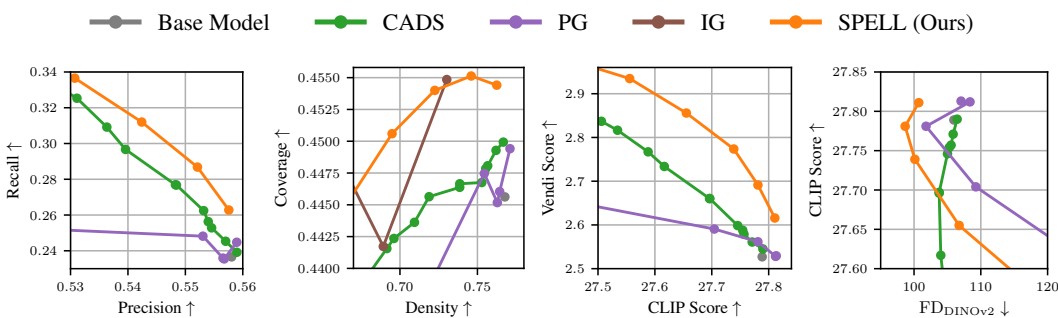

Figure 4: **Latent Diffusion on CC12M.** The three plots on the left highlight how the hyperparameters of diversity methods trade off image quality (x-axes) and diversity metrics (y-axes). SPELL provides a better trade-off than other concurrent approaches. In the rightmost plost, highlighting 2 quality metrics, SPELL also shines. IG is not visible on all plots as it strongly decreases image quality.

can counter-intuitively be decreased by more diverse outputs if the diversity takes different forms or is higher than in the reference dataset. This stands out most for SD3, which was not trained on the reference datasets ImageNet-1k/CC12M. We find that SPELL correctly helps SD3 generate images that are generally more diverse, as evidenced by the 26% and 37% increases in the reference-dataset-free Vendi score, but in other attributes than in the reference datasets, explaining the decrease in coverage. Out of the six experiments, precision and density decrease very slightly in three of them, increase for one, and decrease more clearly in 2 (when using SD3). This tradeoff between diversity and precision is common in the literature (Kynkäänniemi et al., 2024; Sadat et al., 2024; Corso et al., 2024), and we show in the next section that SPELL provides more favorable Pareto fronts than alternative recent methods. This tradeoff improves the overall FD$_{\text{DINOv2}}$ score considerably in Latent Diffusion and MDTv2, while staying within 3% of the original value on Simple Diffusion and EDMv2, and increasing on SD3. Overall, we find that SPELL increases the diversity considerably across all models, with only minor tradeoffs in precision.

## 4.3 COMPARISON TO OTHER DIVERSITY-INDUCING METHODS

We now take a closer look at SPELL's hyperparameter, the repellency radius $r$. Figure 3 shows that we can use repellency in two ways. A low radius ($r = 15$) increases the diversity (Vendi score and recall) without compromising on precision. Intuitively, such a small radius only serves to prevent generating the nearly-same image twice. The resulting output set also better reflects the true image distribution, yielding an enhanced FD$_{\text{DINOv2}}$. The second option is to further increase the repellency radius, which as Figure 3 shows further enhances diversity at the cost of precision. There are other current proposals in the literature that enable to control this trade-off. Namely, Interval Guidance (Kynkäänniemi et al., 2024, IG) only applies CFG in a limited time interval in the middle of the backward diffusion. Condition-annealed diffusion sampling (Sadat et al., 2024, CADS) noises the text or class condition that guides the CFG, lowering the noise in later timesteps. Closer in spirit to our proposal, particle guidance (Corso et al., 2024, PG) adds a gradient potential to the backward diffusion at every timestep, such that the diversity of a batch as a whole is increased. We reimplement

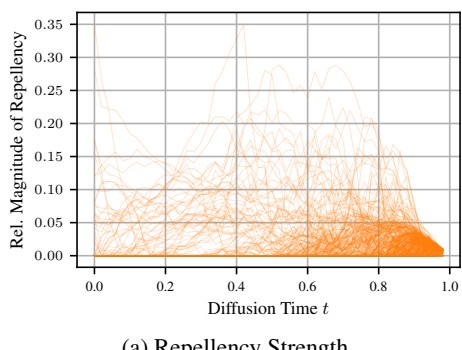
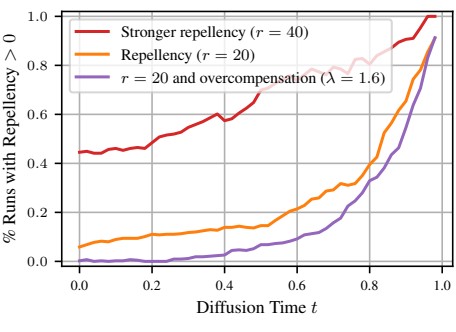

(a) Repellency Strength
(b) Diffusion steps with repellency

Figure 5: (a) The gradient that SPELL adds is only a fraction of the magnitude of the diffusion's score, thus adjusting it without drowning it out. (b) Repellency happens primarily in early backwards steps ($t \in [0.6, 1.0]$) and then often remains zero, thus making it sparse. Overcompensation allows finishing the repellency earlier, whereas runs with high repellency radius repel longer. Latent Diffusion trajectories with repellency from both previous generations and intra-batch.

and tune all approaches and compare them against SPELL. Figure 4 shows that SPELL achieves a more favorable trade-off curve than each of these methods, in three different diversity vs quality Pareto fronts, namely recall vs precision, coverage vs density, and Vendi score vs CLIP Score, as well as a quality/quality plot displaying CLIP/FD$_{DINOv2}$. One reason for this is SPELL's sparsity. SPELL only intervenes if images are going to be too similar to one another. Due to it's ReLU weighting, it remains exactly zero if an image's trajectory is already headed to a diverse output. This leads to increased diversity while leaving high-precision images unchanged. We study this sparsity further in the following section.

### 4.4 SPARSITY ANALYSIS

We focus in this section on the dynamics of SPELL interventions and investigate when and how SPELL corrective terms arise. Figure 5a tracks the magnitude of the SPELL correction vector $\|\Delta\|_2$ normalized by that of the diffusion score vector $\|\nabla \log p_t(x_t|y)\|_2$. We track this relative magnitude throughout 452 backward trajectories for 50 prompts of CC12M with both intra- and inter-batch repellency (Equations (5) and (8)). Appendix H adds further setups. First, we find that repellency only adds a small correction term in most cases. Its magnitude is most often less than 5% of the magnitude of the diffusion score and never exceeds 35%. This explains why our repellency does not reduce image quality or introduce artifacts. A second reason for this is that SPELL corrections happen mostly in the early stages of the backward diffusion, which literature claims to be when the rough image is outlined, rather than in late steps, where the image is refined (Biroli et al., 2024; Kynkäänniemi et al., 2024). Recall that the backwards diffusion starts at $t = 1$ and outputs the final image at $t = 0$. Figure 5b shows that at $t = 0.8$, only 40% of the trajectories have a non-zero repellency term anymore, further declining to 21% at $t = 0.6$. If we impose a higher repellency radius, the repellency acts for longer. Especially in this cases, adding overcompensation helps. As intended, the repellency strength is increased and in return stops earlier. These stops are often final: The repellency stays zero for the remainder of the generation, verifying that the trajectories do not bounce back into the repellency radii, as shown in Appendix H, and reaffirming SPELL's sparsity.

### 4.5 QUALITATIVE EXAMPLE OF SPELL INTERVENTIONS

Figure 6 shows 16 images generated iteratively using SD3 with and without SPELL interventions. Images are generated one by one ($B = 1$), and when generating the $i + 1$-th image, SPELL repels from the reference set of all images 1 to $i$ it has generated thus-far. Images are highlighted in orange if SPELL enforced changes during their generation trajectory. When SPELL does not intervene, the SD3 + SPELL image (bottom row) coincides with the SD3 output (top row), since we use the same seeds. For images 1 to 3, SPELL did not intervene as it did not detect similarities when generating the 2nd image when compared to the 1st, nor when generating the 3rd w.r.t. 1st and 2nd. The 4th image was expected to come out too close to the 3rd at some time during generation, triggering

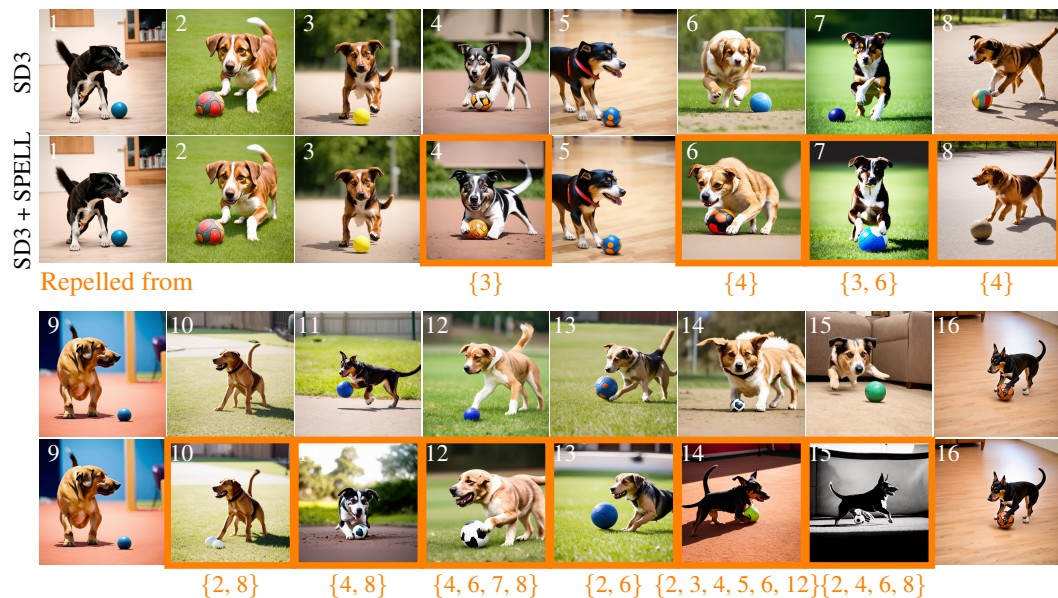

Figure 6: Images for the prompt *"a dog plays with a ball"* generated one-by-one ($B = 1$) with SD3 (top) and SD3 + SPELL (bottom), using the same random seeds. SPELL intervenes on 10 of the 16 images, marked with orange borders, because they are too similar to previously generated images. Early images are adjusted less often and mostly in details because they are still novel enough. Later images repel from more previous images and more strongly to ensure they are different enough.

SPELL to alter the ball color, dog, and background. As more images are added to the reference set, SPELL intervenes more often. For example, image 14 is headed towards an image of a dog on a grassy ground with trees in the background, which is too similar to multiple previous images. Hence, SPELL intervenes and leads the diffusion trajectory to an entirely new mode, with both a new dog race and a previously unseen tartan surface. A similarly strong intervention happens in image 15. Still, SPELL's sparsity means that it not intervene if it does not have to, even when there are already many shielded images. This is the case for image 16, which is novel enough to remain unchanged.

### 4.6 IMAGE PROTECTION BENCHMARK

We scale SPELL to a repellency set of 1.2 million ImageNet-1k train images via approximate nearest neighbor search (Douze et al., 2024), with the goal of generating novel images. We generate 50k images, track how often they fall into a protected shield, as well as the precision, recall, and runtime. This is similar to machine unlearning for generative models (Wang et al., 2024a; Liu et al., 2024), with the main differences that we wish to shield a set of specific images rather than global concepts (Wu et al., 2024; Park et al., 2024), and that SPELL is a training-free intervention.

Table 2 shows that 7.6% of the 50k images that MDTv2 generates without SPELL are within an $L_2$ distance of $r = 60$ of their nearest neighbor on ImageNet. Figure 7 shows examples and verifies that such images indeed are nearly copies of existing images. Adding SPELL with $r = 60$ reduces this rate down to 0.16%. Figure 7 shows that the images are indeed changed in such a way that they are not too close to their ImageNet neighbors anymore. Searching for shields in less Voronoi cells of the approximate nearest neighbor algorithm allows to speed up the generation time, at the cost of a catching less shields. This runtime, and the fact that there are still images within the shields, can largely be attributed to the approximate next neighbor search algorithm over the $K = 1.2M$ images. Appendix G shows that in the diversity experiments with smaller protection sets of $K \leq 128$, SPELL does not increase runtime. Further improvements in $L_2$ based neighbor search techniques will further increase SPELL's protection rate and compute overhead. Expectedly, the recall decreases when repelling from all training images, because validation images may fall into the shield radius around train images. However, the precision remains largely unaffected. This demonstrates again that SPELL does not introduce visual artifacts, even when repelling from many

Table 2: SPELL almost always generates images outside the shields of the ImageNet-1k train set. Searching for shields in more Voronoi cells during the nearest neighbor lookup improves the protection rate at a higher inference-time runtime. The runtime is reported on a single A100-40GB GPU.

| Model | Searched cells | Generated images too close to ImageNet neighbors ↓ | Precision ↑ | Recall ↑ | Time per image (s) ↓ |
|---|---|---|---|---|---|
| EDMv2 | | 7.60% | 0.792 | 0.242 | 2.434 |
| + SPELL | 1 | 1.08% | 0.792 | 0.181 | 4.633 |
| + SPELL | 2 | 0.55% | 0.788 | 0.175 | 6.057 |
| + SPELL | 3 | 0.33% | 0.777 | 0.162 | 7.790 |
| + SPELL | 5 | 0.22% | 0.771 | 0.163 | 9.949 |
| + SPELL | 10 | 0.16% | 0.768 | 0.160 | 13.545 |

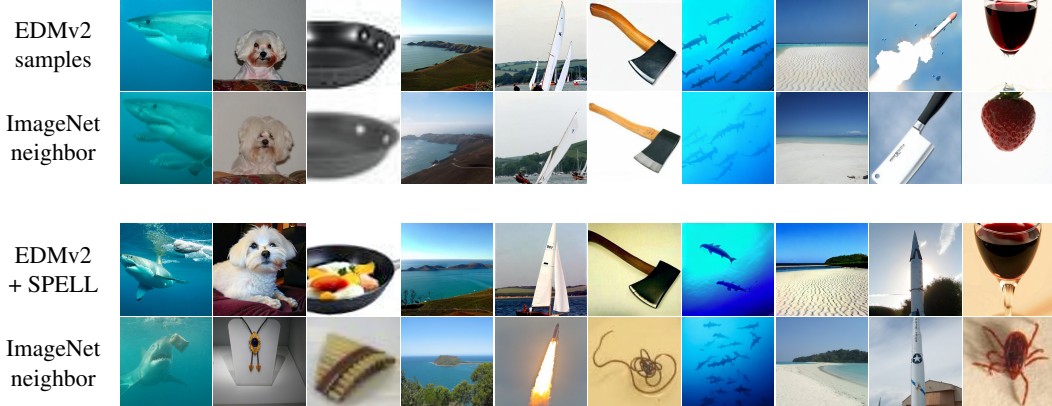

EDMv2 samples

ImageNet neighbor

EDMv2 + SPELL

ImageNet neighbor

Figure 7: The images generated by EDMv2 in the first row are too close to existing images in the ImageNet-1k train set, which EDMv2 was trained on. SPELL adapts their diffusion trajectories to ensure that they maintain a protection radius. The images in third row, generated from the same seeds but with SPELL, are sufficiently different from their nearest neighbor.

images. To add qualitative evidence to this point, we show a random set of images where SPELL intervened in Appendix J. Finally, the last two images in Figure 7 give more insight into the workings of the $L_2$ similarity in the VAE latent space that MDTv2 diffuses in. Apparently, image distances inside the VAE space encode a visual similarity where images with similar colors and compositions are close to one another. SPELL could also create shields in semantic spaces, e.g., by comparing the DINOv2 embeddings of expected image outputs, which we leave for future works.

## 5 DISCUSSION

This work introduces sparse repellency (SPELL), a training-free post-hoc method to guide diffusion models *away* from a certain set of images. This both prevents repeating images that were already generated, thereby increasing diversity, and allows protecting a certain set of reference images, with applications like machine unlearning. SPELL can be applied to any diffusion model, whether it is class-to-image or text-to-image, and whether it is unconditional or classifier(-free) guided. We see two main ways in how future work can extend SPELL. First, SPELL can currently be proven to guarantee to generate images outside the shields if all shields are disjoint. If there are overlapping shields, there can be cases where the guarantee is not strict anymore, and the algorithm could be improved to find a direction that points out of the convex hull of all points, at the cost of decreased scalability. Second, we currently apply SPELL with respect to the $L_2$ distance inside the diffusion model's latent spaces, which lead to *visually* different outputs. Using a distance inside a semantically structured space could lead to generating more *semantically* different images. This has the potential to further increase metrics like FID and recall, which evaluate generated images in semantic spaces.

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

## A GUIDANCE VIA DOOB'S H-TRANSFORM

Doob's h-transform provides a definitive approach to conditioning and guiding diffusions. In the context of avoiding points let $S = \cup_k B_k =$ where $B_k = \{x \in \mathcal{X} \ : \ \|x - z_k\|_2 \leq r\}$ are balls of radius $r$ around centers $(z_k)_k$. We may approximate Doob's h transform with some simplifying assumptions based on diffusion posterior sampling (DPS (Chung et al., 2023)). DPS entails approximating $p_{0|t}(x_0 \mid x_t)$ with $\tilde{p}_0(|D(x_t))$, where $D(x_t) = \mathbb{E}[\mathbf{X}_0 \mid x_t]$ for some choice of density $\tilde{p}$.

Let us observe that

$$\nabla \log p_{0|t}(\mathbf{X}_0 \notin \cup_k B_k \mid x_t) = \sum_k \nabla \log p_{0|t}(\mathbf{X}_0 \notin B_k \mid x_t).$$

For simplicity, we approximate the conditional density $p_{0|t}(\cdot \mid x_t)$ with a Gaussian density with mean $\mathbb{E}[\mathbf{X}_0 \mid x_t]$ and variance $\Sigma_t = \mathrm{Id}$. Since, for $X \sim \mathcal{N}(\mu, \mathrm{Id})$, the random variable $\|X - z_k\|^2$ follow a non-centered chi-square distribution $\chi^2_{nc,d}(\lambda)$ where $\lambda := \lambda(\mu) = \|\mu - z_k\|^2$. As such

$$\nabla \log p_{0|t}(\mathbf{X}_0 \notin B_k \mid x_t) \approx \frac{-\nabla_\mu F_{\chi^2_{nc,d}(\lambda)}(r^2)}{1 - F_{\chi^2_{nc,d}(\lambda)}(r^2)} = (\mu - z_k) \times \omega(\lambda(\mu), r)$$

with weights function

$$\omega(\lambda, r) = \frac{2}{F_{\chi^2_{nc,d}(\lambda)}(r^2) - 1} \times \frac{\partial F_{\chi^2_{nc,d}(\lambda)}}{\partial \lambda}(r^2).$$

We recall that the CDF of $\chi^2_{nc,d}$ is a combination of the CDF of standard $\chi^2_d$:

$$F_{\chi^2_{nc,d}(\lambda)}(x) = \sum_{j=0}^\infty c_j(\lambda) F_{\chi^2_{d+2j}}(x)$$

$$\frac{\partial F_{\chi^2_{nc,d}(\lambda)}}{\partial \lambda}(x) = \frac{1}{2} \sum_{j=0}^\infty c_j(\lambda) \left[ F_{\chi^2_{d+2(j+1)}}(x) - F_{\chi^2_{d+2j}}(x) \right]$$

$$\nabla_\mu F_{\chi^2_{nc,d}(\lambda)}(x) = \frac{\partial F}{\partial \lambda}(x) \times \partial_\mu \lambda(\mu) = \frac{\partial F}{\partial \lambda}(x) \times 2(\mu - z_k)$$

where we denoted $c_j(\lambda) = \frac{(\lambda/2)^j e^{-\lambda/2}}{j!}$.

## B  REPELLENCE GUARANTEE

Consider again our adjusted SDE

$$dx = \left[ f(x,t) - g^2(t) \left( \nabla_x \log p_t(x) + \frac{\alpha_t}{\sigma_t^2} \sum_{k=1}^K (\hat{x}_0 - z_k) \mathrm{ReLU} \left( \frac{r}{||\hat{x}_0 - z_k||_2} - 1 \right) \right) \right] dt + g(t) dw,$$

where $\hat{x}_0 := \frac{x_t + \sigma_t^2 \nabla_x \log p_t(x)}{\alpha_t}$ is the expected image if we did not intervene.

This section shows that the SDE leads to a output distribution with $P_0(\{B_r(z_k)|k=1,\ldots,K\}) = 0$. This ensures that it does not create samples within radius $r$ around the repellence images $z_k, k = 1, \ldots, K$. To this end, assume we have a set of repellence images and that their repellence balls do not overlap (otherwise, one can merge them and select an according higher radius). Let's consider an arbitrary timestep $t$. Then Tweedie's formula (Efron, 2011; Bradley & Nakkiran, 2024) gives that

$$\mathbb{E}[\mathbf{X}_0|x_t] = \frac{x_t + \sigma_t^2 \left( \nabla_x \log p_t(x) + \frac{\alpha_t}{\sigma_t^2} \sum_{k=1}^K (\hat{x}_0 - z_k) \mathrm{ReLU} \left( \frac{r}{||\hat{x}_0 - z_k||_2} - 1 \right) \right)}{\alpha_t}$$

$$= \frac{x_t + \sigma_t^2 \nabla_x \log p_t(x)}{\alpha_t} + \frac{\sigma_t^2}{\alpha_t} \frac{\alpha_t}{\sigma_t^2} \sum_{k=1}^K (\hat{x}_0 - z_k) \mathrm{ReLU} \left( \frac{r}{||\hat{x}_0 - z_k||_2} - 1 \right)$$

$$= \hat{x}_0 + \sum_{k=1}^K (\hat{x}_0 - z_k) \mathrm{ReLU} \left( \frac{r}{||\hat{x}_0 - z_k||_2} - 1 \right)$$

**Case 1:** $||\hat{x}_0 - z_k||_2 \geq r \, \forall k = 1, \ldots, K$. Then the ReLU term becomes 0 and $\hat{x}_0$ remains unadjusted and $||\mathbb{E}[\mathbf{X}_0|x_t] - z_k||_2 \geq r$.

**Case 2:** $\exists k^\star \in \{1, \ldots, K\} : ||\hat{x}_0 - z_k^*||_2 < r$. Since the balls are non-overlapping,

$$\sum_{k=1}^K (\hat{x}_0 - z_k) \mathrm{ReLU} \left( \frac{r}{||\hat{x}_0 - z_k||_2} - 1 \right) = (\hat{x}_0 - z_k^*) \mathrm{ReLU} \left( \frac{r}{||\hat{x}_0 - z_k^*||_2} - 1 \right).$$

Then

$$||\mathbb{E}[\mathbf{X}_0|x_t] - z_k||_2 = ||\hat{x}_0 + (\hat{x}_0 - z_k^*) \mathrm{ReLU} \left( \frac{r}{||\hat{x}_0 - z_k^*||_2} - 1 \right) - z_k^*||_2$$

$$= ||(\hat{x}_0 - z_k^*) + (\hat{x}_0 - z_k^*) \left( \frac{r}{||\hat{x}_0 - z_k^*||_2} - 1 \right)||_2$$

$$= ||(\hat{x}_0 - z_k^*) + (\hat{x}_0 - z_k^*) \frac{r}{||\hat{x}_0 - z_k^*||_2} - (\hat{x}_0 - z_k^*)||_2$$

$$= ||r \frac{(\hat{x}_0 - z_k^*)}{||\hat{x}_0 - z_k^*||_2}||_2$$

$$= r$$

So, in all cases, $||\mathbb{E}[\mathbf{X}_0|x_t] - z_k||_2 \geq r$, for any $t$. Especially, for $t = 0$, the SDE does not add any noise anymore and the sampled $x_0$ is equal to the expectation.

Hence $||x_0 - z_k||_2 \geq r \, \forall k = 1, \ldots, K$.

## C  CONSERVATIVE FIELD INTERPRETATION

The function $h(x) = \text{ReLU}(\frac{r}{\|x\|} - 1)x$ is the conservative field associated to the (family of) potential $H : \mathbb{R}^d \to \mathbb{R}$:

$$H(x) = \begin{cases} r\|x\| - \frac{1}{2}\|x\|^2 \text{ when } \|x\| < r, \\ \frac{r^2}{2} \text{ otherwise,} \end{cases} \tag{9}$$

where Gauge $H(0)$ is chosen arbitrarily, as illustrated in Figure 8.

Furthermore, observe that the mapping $x_t \mapsto \hat{x}_0$ defined by $\hat{x}_0 = \frac{1}{\alpha_t}\left(\sigma_t^2 \nabla \log p_t(x_t) + x_t\right)$ is a conservative field given by the potential $\frac{1}{\alpha_t}\left(\sigma_t^2 \log p_t(x_t) + \frac{1}{2}\|x_t\|^2\right)$.

Therefore, SPELL guidance in Equation 5 is the composition of two conservative fields. But note that, in general, conservative fields are not stable by composition, unless the Hessians of their potentials commute everywhere.

**Theorem 1.** *We consider $f : \mathbb{R}^d \to \mathbb{R}$ a twice differentiable function. The Jacobian of the map $\phi : x \mapsto \frac{\nabla f(x)}{\|\nabla f(x)\|}$ is given by*

$$\text{Jac}(\phi)(x) = \frac{1}{\|g\|}H - \frac{1}{\|g\|^3}gg^T H, \text{ with } g = \nabla f(x) \text{ and } H = \nabla^2 f(x)$$

Hence, in the case where $H$ and $gg^T$ commute, this Jacobian is locally symmetric. If they commute everywhere, then this Jacobian is globally symmetric, and $\phi$ is a conservative field.

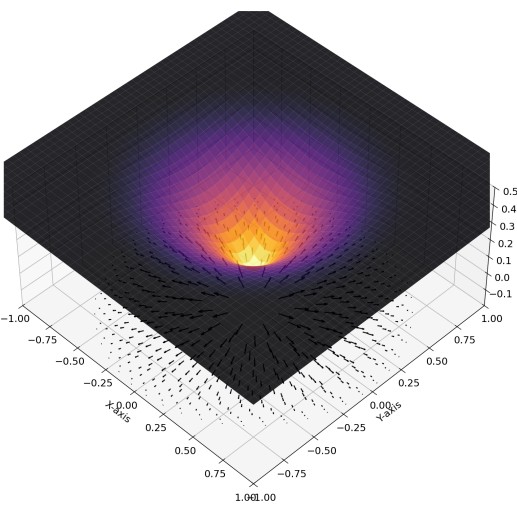

Figure 8: Potential function whose gradient field is $\text{ReLU}(\frac{r}{\|x\|} - 1)x$, displayed for $x \in \mathbb{R}^2$. Repellence force is dynamic: closer to the center (i.e., when a diffusion trajectory is expected to be close to a protected image) it applies stronger gradients, as shown by the arrows, while outside the repellecy radius, it applies no gradient at all, letting the diffusion trajectory continue without any intervention.

## D  IMPLEMENTATION DETAILS AND HYPERPARAMETERS

Since SPELL is a training-free post-hoc method, we use the trained checkpoints of diffusion models provided by their original authors. For EDMv2 and MDTv2, we use the hyperparameters suggested by their authors. Latent Diffusion, Simple Diffusion, and Stable Diffusion come without recommended hyperparameters, so we tune the classifier-free guidance (CFG) weight by the F-score between precision and coverage on the 554 validation captions on our CC12M split.

For the repellence radius $r$, the latent spaces that the different models diffuse on have different dimensionalities, hence the scales of the repellence radii differ. To get a sense of the scales, we first generate one batch of images without repellence and tracked the pairwise $L_2$ distance between generated latents at the final timestep. We then test 16 values from 0 to 2 times the median distance. This yields the following hyperparameters for the results in Table 1.

**EDMv2:** CFG weight 1.2, 50 backwards steps, $\sigma_{\min} = 0.002$, $\sigma_{\max} = 80$, $\rho = 7$, $S_{\min} = 0$, $S_{\max} = \infty$, repellence radius $r = 20$, batchsize 8.

**MDTv2:** CFG weight 3.8, 50 backwards steps, repellence radius $r = 45$, batchsize 2.

**Stable Diffusion 3:** CFG weight 5.5, 28 backwards steps, repellence radius $r = 200$, on CC12M overcompensation 1.6 (no overcompensation on ImageNet), batchsize 8.

**Simple Diffusion:** CFG weight 5.5, 50 backwards steps, repellence radius $r = 50$, overcompensation 1.6, batchsize 16.

**Latent Diffusion:** CFG weight 5, 50 backwards steps, repellence radius $r = 20$, overcompensation 1.6, batchsize 8.

Algorithm 1 gives a high-level pseudo-code for SPELL and Algorithm 2 details how we implemented SPELL in a parallelized way in Python.

---

**Algorithm 1:** SPELL added to the backwards diffusion step. This is a simplified example, see Appendix D for Python code that is parallelized and supports sparse neighbor retrieval.

---

1 **Input:** *Batch of latents* $\{x_t^{(b)}\}_{b=1,\ldots,B}$, *set of shielded images* $\{z_k\}_{k=1,\ldots,K}$, *radius* $r$, $\lambda$
2 **for** $b = 1, \ldots, B$ **do**
3     $\hat{x}_0^{(b)} \leftarrow D_{\theta^\star}(t, x_t^{(b)})$            ▷ *Expected diffusion output without repellency*
4 **end**
5 **for** $b = 1, \ldots, B$ **do**
6     $\vec{\Delta}_b \leftarrow 0$
7     **for** $k = 1, \ldots, K$ **do**            ▷ *Repel from the shielded set*
8        **if** $\|\hat{x}_0^{(b)} - z_k\|_2 < r$ **then**
9           $\vec{\Delta}_b = \vec{\Delta}_b + (\hat{x}_0^{(b)} - z_k) \operatorname{ReLU}\left(\frac{r}{\|\hat{x}_0^{(b)} - z_k\|_2} - 1\right)$
10        **end**
11     **end**
12     **for** $b' = 1, \ldots, B, b' \neq b$ **do**            ▷ *Repel within the batch*
13        **if** $\|\hat{x}_0^{(b)} - \hat{x}_0^{(b')}\|_2 < r$ **then**
14           $\vec{\Delta}_b = \vec{\Delta}_b + (\hat{x}_0^{(b)} - \hat{x}_0^{(b')}) \operatorname{ReLU}\left(\frac{r}{\|\hat{x}_0^{(b)} - \hat{x}_0^{(b')}\|_2} - 1\right)$
15        **end**
16     **end**
17     Calculate $x_{t-1}^{(b)}$ by taking a step towards $\hat{x}_0^{(b)} + \lambda\vec{\Delta}_b$ (using the diffusion scheduler)
18 **end**
19 **Output:** $\{x_{t-1}^{(b)}\}_{b=1,\ldots,B}$

---

```python
def backward_step(x_t, t, protection_set, r, lambda, repel_within_batch):
    """
    A generation step from t to t-1 of a diffusion with repellency.

    x_t: Matrix of size [batch, dimensions] containing the current latents
    t: float, current time
    protection_set: Either a matrix of size [num_protection_images, dimensions] with latents
     we want to repel from or a database that will output closest neighbors in this format
    radius: Float, repellency radius
    lambda: Float, overcompensation factor
    repel_within_batch: Boolean, whether to apply intra-batch repellency
    """

    # Predict x_0 using the diffusion model (using diffusion without repellency)
    x_0_hat = diffusion_score.predict(x_t, t)

    # Repel from protection set
    repellency_term = 0
    if protection_set is not None:
        if isinstance(protection_set, database):
            protection_set = protection_set.find_neighbors_within_radius(x_0_hat, radius)
        diff_vec = x_0_hat.unsqueeze(1) - mu.unsqueeze(0)
        # diff_vec has size [batch, num_protection_images, dimensions]
        weight = (diff_vec**2).sum(dim=2).sqrt()
        trunc_weight = ReLU(radius / diff - 1)
        repellency_term += (diff_vec * trunc_weight).sum(dim=1)

    # Repel within batch
    if repel_within_batch:
        diff_vec = x_0_hat.unsqueeze(1) - x_0_hat.unsqueeze(0)
        # diff_vec has size [batch, batch, dimensions]
        weight = (diff_vec**2).sum(dim=2).sqrt()
        trunc_weight = ReLU(radius / diff - 1)
        diag(trunc_weight) = 0  # Don't repel from the image itself
        repellency_term += (diff_vec * trunc_weight).sum(dim=1)

    # Add our repellency term to the current x_0_hat prediction
    x_0_hat = x_0_hat + lambda * repellency_term

    # Step from t to t-1 using the diffusion update rule (same as in typical diffusion)
    x_t_minus_1 = calculate_update(x_0_hat, x_t, t)
    if t > 0:
        x_t_minus_1 += generate_noise(t)

    return x_t_minus_1
```

Algorithm 2: Our repellency can be added to the backwards algorithm of existing diffusion models, without retraining. Since the expected x_0_hat is often already computed as part of the backward process, the only runtime overhead are the pairwise differences and the possible neighbor search.

## E  CONSTRUCTION OF THE SOFT-LABEL CC12M DATASET

CC12M is a recent text-to-image dataset that contains pairs of image links and the title scraped from their metadata. To turn this into our soft-label subset of CC12M, where each caption has a set of multiple possible images related to it, we first group all images in CC12M by their caption and keep only captions with at least four and at most 128 images.

Some of these images are falsely grouped together. For example, there are photo albums whose images were assigned the same generic title in their metadata. A useful heuristic to filter out such cases is to analyze the top-level domains of the images. We filter out sets where the most frequent top-level domain belongs to $75\%$ or more of the image urls. Second, we filter out automatically generated captions by removing captions that include the strings 'Display larger image', 'This image may contain', 'This is the product title', or 'Image result for'. Last, due to privacy guidelines, we filter out any caption whose image may include individuals. This is done by filtering out caption that include '<PERSON>', which is a placeholder that the CC12M dataset overwrote any possible person name with. After these filtering steps, we arrive at 5554 captions. We randomly split them into a validation set of 554 captions and a test set of 5000 captions. Table 3 shows how many images belong to each caption.

We did not filter any images out although there are some near-duplicates. This is done on purpose in order to not skew the distributions. Filtering out captions amounts to deciding on which subset of the dataset we test our models on. But filtering out images would change the conditional distributions $P(X|c)$ to something different from the training distributions. In other words, a model that learned the train distribution ideally is expected to have a stronger mode at near-duplicate images but testing it on a changed $P(X|c)$ distribution would punish it for learning the correct distribution. If a future work intends to test models on unseen images, we note that removing near-duplicates may be a possibility, depending on the experiment design.

Table 3: Number of captions that have a certain number of images attached to them in our soft-label CC12M dataset.

| Images per caption | Validation split | Test split |
| --- | --- | --- |
| $4-5$ | 270 | 2600 |
| $6-10$ | 174 | 1485 |
| $11-20$ | 75 | 555 |
| $21-30$ | 20 | 219 |
| $31-40$ | 10 | 86 |
| $41-50$ | 3 | 32 |
| $51-128$ | 2 | 23 |

## F  FURTHER DIVERSITY-QUALITY TRADEOFFS

In addition to the tradeoff experiments in Section 4.3, Table 4 provides the full combinations of metrics attainable with each method, depending on how one chooses the hyperparameters. This is the raw data underlying Figure 4 and allows the curious reader to compare arbitary tradeoffs.

## G  RUNTIME ANALYSIS AND COMPARISON

The scale of the overhead that SPELL adds is negligible when contrasted with the diffusion generation cost. It amounts to computing (up to) $[B, K]$, distance matrices per time $t$, where both the batchsize $B$ and the size of the protection set $K$ do not exceed hundreds, and adding one single correction vector to the score. Table 5 confirms that the runtime that SPELL adds (as well as the other benchmarked diversity methods) is negligible, here using $B = 8$ and intra-batch repellency, hence $K = B - 1 = 7$. This also further confirms that the runtime observed in Section 4.6 is due to the next-neighbor search algorithm, not SPELL's correction terms.

## H  ABLATION: REPELLENCY STRENGTH THROUGHOUT THE GENERATION

In this section, we scrutinize how and when repellency acts during the generation. We also use these insights to run ablations that foster the intuition on the role of the repellence radius.

To begin with, Figure 9 shows repellency in the standard setting with a repellency radius of 25 in Latent Diffusion. We first generate 8 images per prompt, and then generate another 8 images that repel from the first ones, without intra-batch repellency. Figure 9a shows how high the $L_2$ norm of the total gradient is that our repellency adds to the score, divided by the $L_2$ norm of the score. It can be seen that the repellency term is in most cases at most $20\%$ as strong as the original diffusion gradient field. Intuitively, this means that our repellence does not drown out the diffusion model, but is more a corrective term. Repellency mostly takes place early in the backwards diffusion ($t \in [0.6, 1.0]$), with Figure 9b demonstrating that more than $50\%$ of the generations have already finished their repellency in the first quarter of timesteps (note that Latent Diffusion uses linearly scheduled timesteps). This leaves sufficient time for the diffusion model to generate high quality images in the remainder of steps.

Figure 10 uses intra-batch repellency instead of repelling from 8 previously generated images. The dynamics are very similar to Figure 9 (see also the comparison in Figure 15). This shows that our

Table 4: Metrics of all approaches in the tradeoff experiments in Figure 4.

| Method | Recall | Vendi Score | Coverage | Precision | Density | FID | $FD_{\text{DINOv2}}$ | CLIP Score |
|---|---|---|---|---|---|---|---|---|
| Base Model | 0.237 | 2.527 | 0.446 | 0.558 | 0.768 | 9.566 | 105.967 | 27.789 |
| Particle Guidance, strength = 1024 | 0.099 | 1.987 | 0.249 | 0.300 | 0.326 | 84.115 | 705.661 | 24.470 |
| Particle Guidance, strength = 512 | 0.230 | 2.753 | 0.378 | 0.443 | 0.534 | 23.106 | 286.093 | 26.740 |
| Particle Guidance, strength = 256 | 0.252 | 2.656 | 0.429 | 0.523 | 0.682 | 11.934 | 154.897 | 27.440 |
| Particle Guidance, strength = 128 | 0.248 | 2.591 | 0.447 | 0.553 | 0.754 | 9.442 | 109.257 | 27.704 |
| Particle Guidance, strength = 64 | 0.245 | 2.561 | 0.449 | 0.559 | 0.771 | 9.072 | 101.796 | 27.781 |
| Particle Guidance, strength = 32 | 0.235 | 2.528 | 0.445 | 0.557 | 0.763 | 9.724 | 108.382 | 27.812 |
| Particle Guidance, strength = 16 | 0.236 | 2.529 | 0.446 | 0.557 | 0.764 | 9.596 | 107.041 | 27.813 |
| Interval Guidance, [0.1,0.9] | 0.372 | 2.840 | 0.455 | 0.537 | 0.730 | 8.385 | 85.871 | 27.453 |
| Interval Guidance, [0.2,0.9] | 0.419 | 2.994 | 0.442 | 0.514 | 0.689 | 8.359 | 85.094 | 26.813 |
| Interval Guidance, [0.1,0.8] | 0.470 | 3.174 | 0.448 | 0.500 | 0.663 | 7.507 | 76.104 | 27.215 |
| Interval Guidance, [0.3,0.9] | 0.471 | 3.208 | 0.421 | 0.483 | 0.635 | 8.406 | 87.971 | 25.885 |
| Interval Guidance, [0.2,0.8] | 0.518 | 3.340 | 0.434 | 0.478 | 0.624 | 7.478 | 75.250 | 26.544 |
| Interval Guidance, [0.1,0.7] | 0.567 | 3.576 | 0.432 | 0.451 | 0.577 | 6.804 | 72.092 | 26.784 |
| Interval Guidance, [0.4,0.9] | 0.525 | 3.495 | 0.395 | 0.442 | 0.569 | 8.623 | 96.611 | 24.630 |
| Interval Guidance, [0.3,0.8] | 0.571 | 3.575 | 0.411 | 0.446 | 0.570 | 7.556 | 78.887 | 25.549 |
| Interval Guidance, [0.2,0.7] | 0.614 | 3.770 | 0.417 | 0.426 | 0.536 | 6.771 | 72.972 | 25.979 |
| Interval Guidance, [0.1,0.6] | 0.673 | 4.138 | 0.396 | 0.385 | 0.466 | 6.885 | 81.643 | 26.020 |
| CADS, mixture factor = 0, $\tau_1$ = 0.6 | 0.262 | 2.598 | 0.447 | 0.553 | 0.753 | 9.248 | 105.006 | 27.746 |
| CADS, mixture factor = 0, $\tau_1$ = 0.7 | 0.253 | 2.579 | 0.448 | 0.555 | 0.757 | 9.288 | 105.549 | 27.757 |
| CADS, mixture factor = 0, $\tau_1$ = 0.8 | 0.245 | 2.561 | 0.449 | 0.557 | 0.762 | 9.356 | 105.856 | 27.771 |
| CADS, mixture factor = 0, $\tau_1$ = 0.9 | 0.239 | 2.545 | 0.450 | 0.559 | 0.767 | 9.452 | 106.455 | 27.790 |
| CADS, mixture factor = 0.001, $\tau_1$ = 0.6 | 0.325 | 2.816 | 0.442 | 0.531 | 0.696 | 8.897 | 105.081 | 27.534 |
| CADS, mixture factor = 0.001, $\tau_1$ = 0.7 | 0.297 | 2.734 | 0.446 | 0.540 | 0.719 | 8.963 | 104.006 | 27.617 |
| CADS, mixture factor = 0.001, $\tau_1$ = 0.8 | 0.277 | 2.660 | 0.447 | 0.548 | 0.739 | 9.098 | 103.766 | 27.697 |
| CADS, mixture factor = 0.001, $\tau_1$ = 0.9 | 0.256 | 2.588 | 0.448 | 0.554 | 0.755 | 9.273 | 105.268 | 27.754 |
| CADS, mixture factor = 0.002, $\tau_1$ = 0.6 | 0.425 | 3.208 | 0.417 | 0.472 | 0.584 | 9.870 | 129.159 | 26.920 |
| CADS, mixture factor = 0.002, $\tau_1$ = 0.7 | 0.380 | 3.028 | 0.429 | 0.501 | 0.637 | 9.143 | 114.333 | 27.242 |
| CADS, mixture factor = 0.002, $\tau_1$ = 0.8 | 0.330 | 2.837 | 0.442 | 0.529 | 0.692 | 8.893 | 105.511 | 27.506 |
| CADS, mixture factor = 0.002, $\tau_1$ = 0.9 | 0.277 | 2.660 | 0.446 | 0.548 | 0.739 | 9.098 | 103.762 | 27.696 |
| SPELL, shield radius = 40 | 0.370 | 2.998 | 0.437 | 0.500 | 0.631 | 13.072 | 140.841 | 27.397 |
| SPELL, shield radius = 35 | 0.359 | 2.935 | 0.445 | 0.518 | 0.665 | 11.452 | 120.346 | 27.556 |
| SPELL, shield radius = 30 | 0.337 | 2.856 | 0.451 | 0.531 | 0.695 | 10.349 | 106.753 | 27.655 |
| SPELL, shield radius = 25 | 0.312 | 2.774 | 0.454 | 0.542 | 0.723 | 9.794 | 100.123 | 27.739 |
| SPELL, shield radius = 20 | 0.287 | 2.691 | 0.455 | 0.552 | 0.746 | 9.535 | 98.666 | 27.781 |
| SPELL, shield radius = 15 | 0.263 | 2.616 | 0.454 | 0.558 | 0.762 | 9.558 | 100.709 | 27.811 |

Table 5: Generation times per image. Neither SPELL nor other diversity inducing methods add considerable runtime. The runtime is dominated by the diffusion backbone. Mean $\pm$ standard deviation across 500 images, run on an NVIDIA V100 GPU.

| Model | Generation time per image (seconds) |
|---|---|
| Baseline (Simple Diffusion) | $2.93 \pm 0.12$ |
| Simple Diffusion + PG | $2.96 \pm 0.13$ |
| Simple Diffusion + IG | $2.93 \pm 0.12$ |
| Simple Diffusion + CADS | $2.96 \pm 0.12$ |
| Simple Diffusion + SPELL | $2.94 \pm 0.13$ |

repellency smoothly can be used both intra-batch or iteratively, or in a mixture of both, to generate arbitrary amounts of diverse data even when GPU memory is limited. This mixed setup is presented in Figure 11, where we generate two images at a time that repel both intra-batch and from the previous images. It behaves similarly in both magnitude and duration of repellency. Figure 12 further investigates scalability. Despite repelling from 64 previously generated images, the repellency magnitudes and times are only slightly increased compared to Figure 9. Note that this is despite generating 64+8 images conditionally on the same prompt, repellency from a dataset of more various images like in Section 4.6 is even less effected.

If repellency needs to protect a large radius, the repellency takes place longer in the backwards diffusion process, as shown in Figure 13, where we use an increased radius of 37.5. Here, 43% of the backwards diffusions apply repellency until the end of the generation. The repellency magnitude is increased but still stays below 50% of the magnitude of the diffusion score. One option to speed up the repellency if it runs until the end like here is overcompensation. Figure 14 shows that compared to Figure 9, the repellency is stronger at start and manages to push the trajectories into the diffusion cones of different modes, in return allowing to stop the repellency earlier. This implies that overcompensation can also be used as a means to realize higher repellency radii, without needing to re-

pel until $t = 0$. We leave this, and possibly expansions with overcompensation or repellency radius schedulers, for future works.

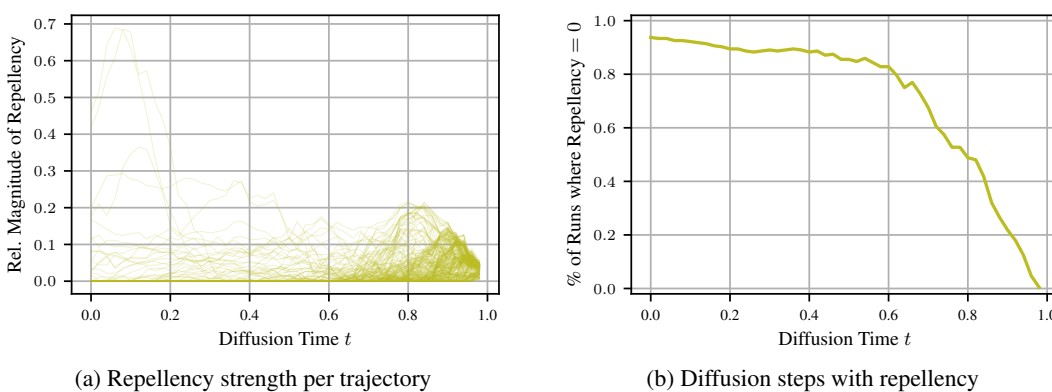

(a) Repellency strength per trajectory

(b) Diffusion steps with repellency

Figure 9: Generating images that repel from 8 protected images (generated with the same prompt). Latent Diffusion, 256 generations in total.

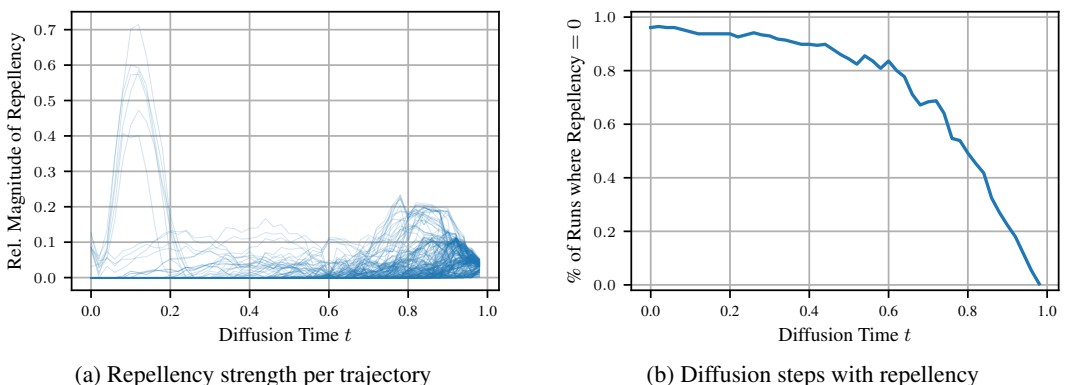

(a) Repellency strength per trajectory

(b) Diffusion steps with repellency

Figure 10: Generating images with the same prompt in batches of 8 with intra-batch repellency. Latent Diffusion, 256 generations in total.

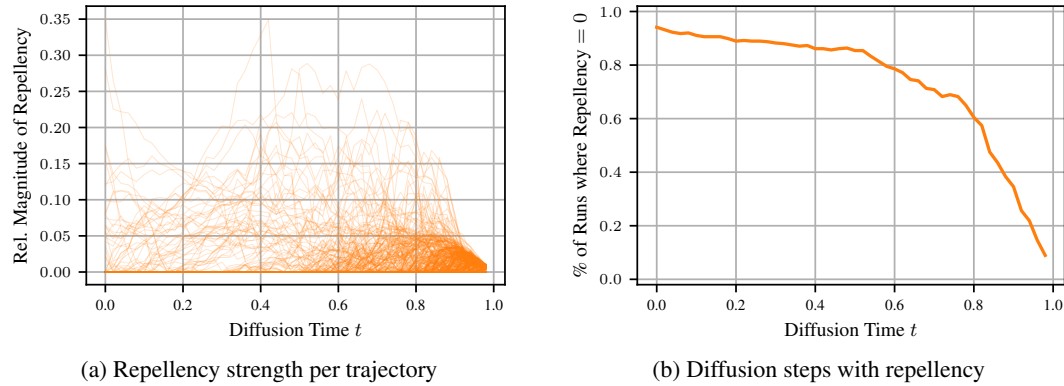

(a) Repellency strength per trajectory

(b) Diffusion steps with repellency

Figure 11: Generating images by iteratively, generating 2 images at a time. They repel both intra-batch and from the previously generated images. We use 50 different prompts, generating 4-32 images each, giving a realistic setup. Latent Diffusion, 452 generations in total.

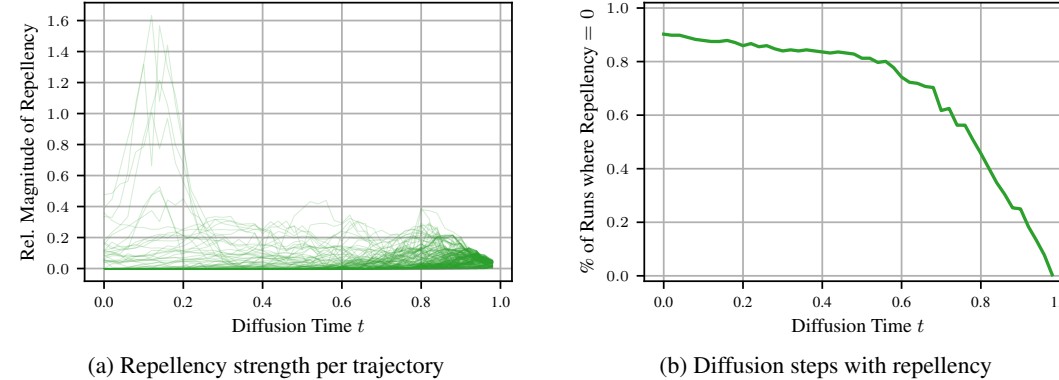

(a) Repellency strength per trajectory

(b) Diffusion steps with repellency

Figure 12: Generating images that repel from 64 protected images (generated with the same prompt). Latent Diffusion, 256 generations in total.

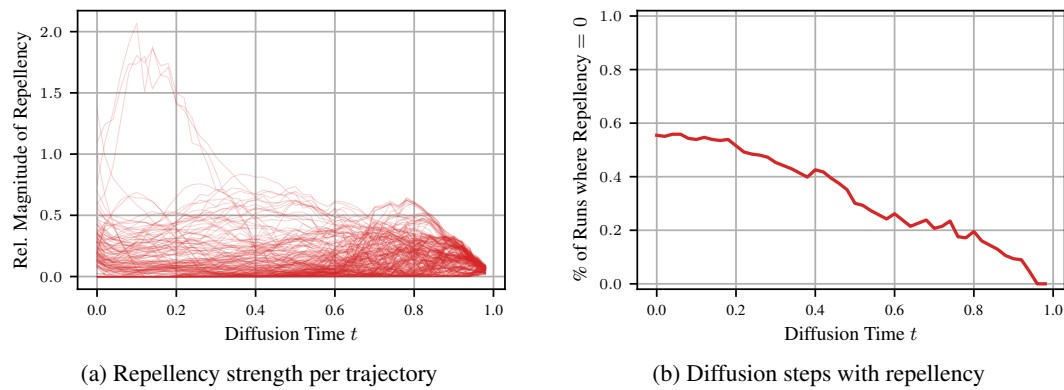

(a) Repellency strength per trajectory

(b) Diffusion steps with repellency

Figure 13: Generating images that repel from 8 protected images (generated with the same prompt), using a 1.5 times larger repellency radius. Latent Diffusion, 256 generations in total.

# I IMAGE PROTECTION ON LARGE DATASETS

Image protection involves computing the repellence between the current batch $x_t$ being generated with a large dataset $\mathcal{D}$ of size $N$, with $N \gg 10^5$. This dataset will be typically too large to fit entirely in GPU memory. Furthermore, computing the repellence term of each element of the batch with every element of the dataset would be prohibitive. However, since the repellence term is zero for vectors that are far-away, this opens the possibility of an optimization: first, the *closest* images from the batch are retrieved using a vector similarity index (stored in RAM), and only then these images are moved into GPU memory for the actual computation of the repellence term. An efficient implementation of this technique is provided by the Faiss library (Douze et al., 2024). We use the IndexIVFFlat object, that rely on Voronoi cells to cluster vectors and speed-up search. We chose a number of Voronoi cells equal to the square root of dataset size, i.e 1131 cells containing typically 1132 examples each. During generation, we probe only the two voronoi cells closest to the current expected outputs. The behavior of the repellence term ensures that false positive are rarely a problem. False negatives (if any) are typically "far-away" which means that their contribution to the sum of all ReLU repellency terms would have been small. In Table 2, we show that one Voronoi cell is often enough. Searching the ten closest cells gives an even higher protection rate, though at the cost of higher searching costs. This shows that advances in efficient search algorithms will directly benefit SPELL when it is applied to large repellency sets.

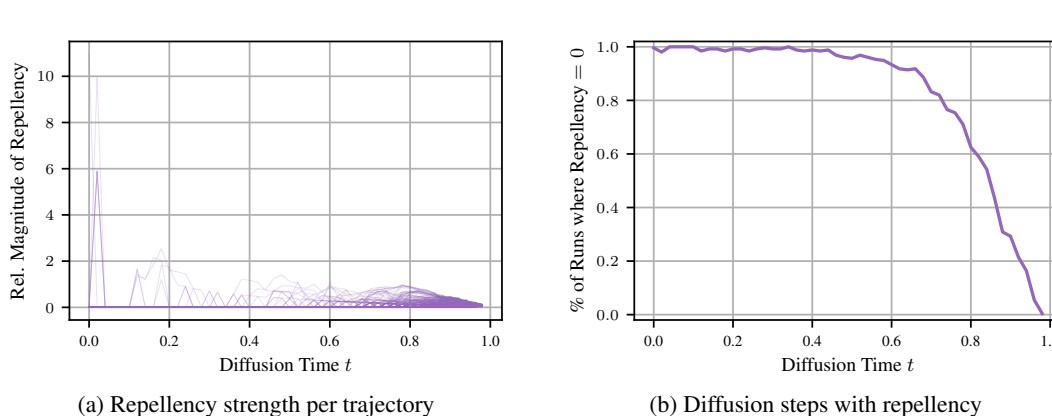

(a) Repellency strength per trajectory

(b) Diffusion steps with repellency

Figure 14: Generating images that repel from 8 protected images (generated with the same prompt), with an overcompensation factor of 2. Latent Diffusion, 256 generations in total.

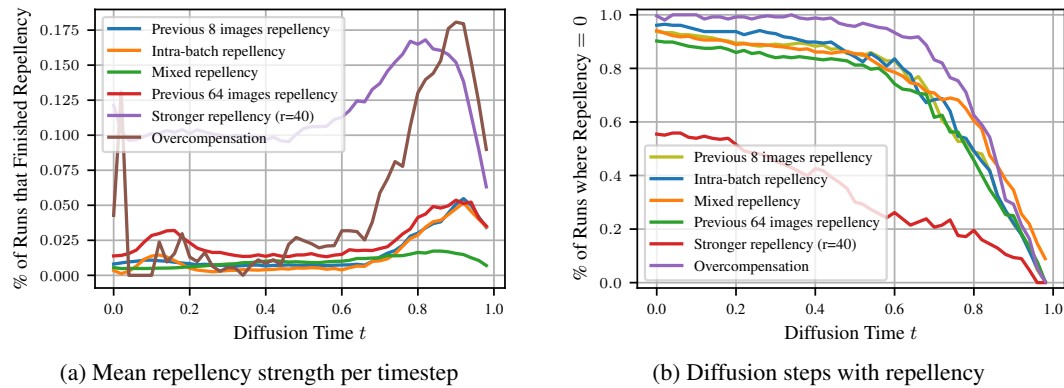

(a) Mean repellency strength per timestep

(b) Diffusion steps with repellency

Figure 15: Comparison of the previous ablations.

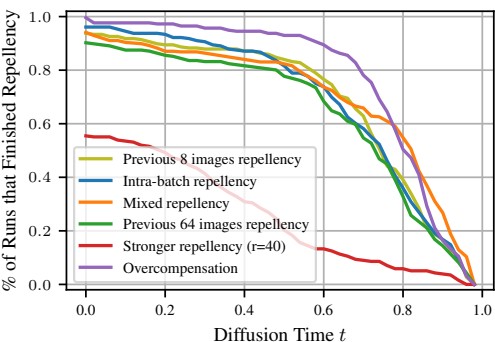

Figure 16: Timesteps at which the repellency has finished, in that the term is zero and stays zero for the remainder of the generation.

## J    EXAMPLES OF IMAGES GENERATED WITH REPELLENCY

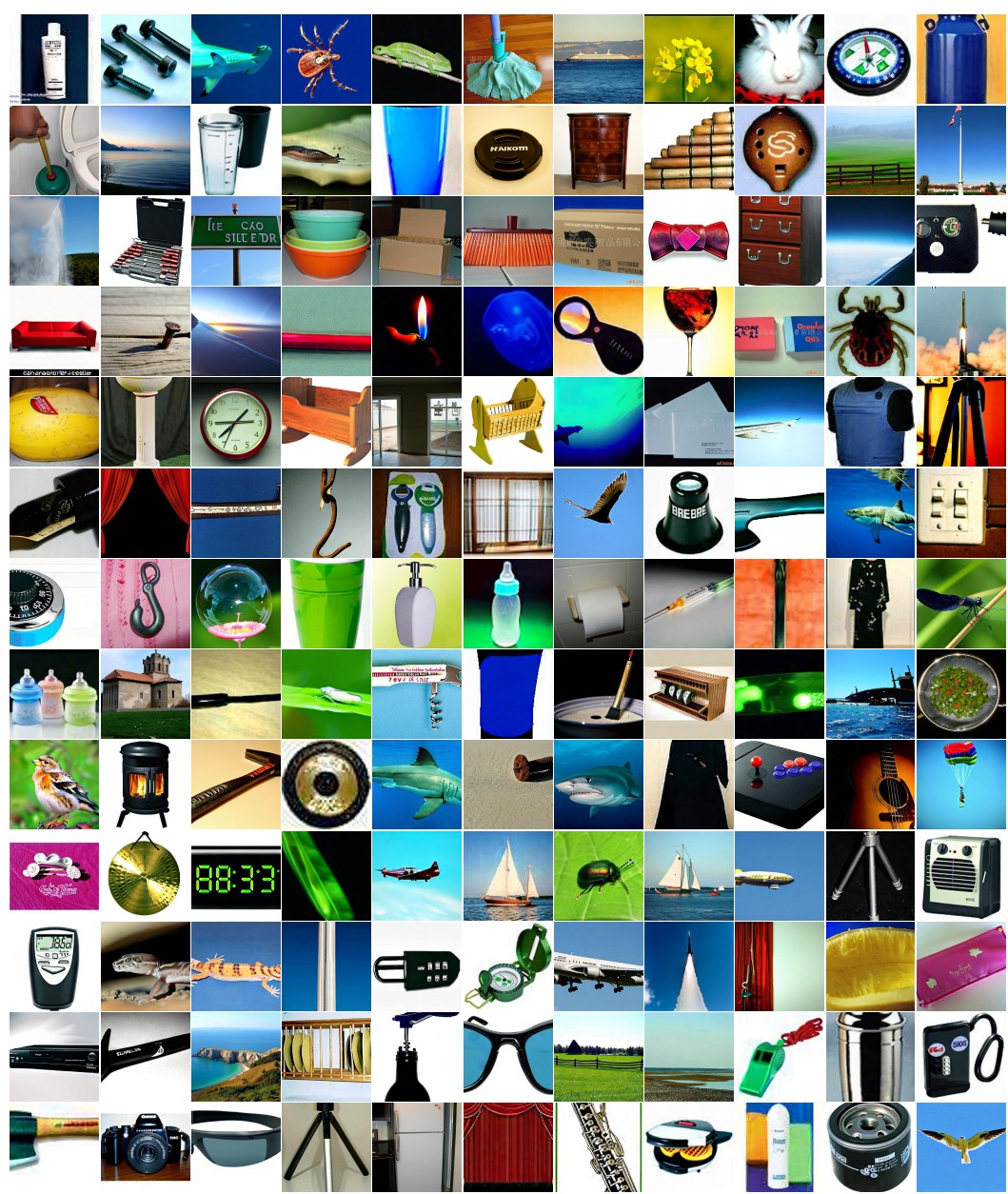

Figure 17: Randomly chosen images where repellency actively pushed EDMv2 away from the protected ImageNet-1k train set in Section 4.6. All images have repellency applied to them but do not show visual artifacts. Low-quality images are by design because the underlying EDMv2 model learned to generate this style of images from the ImageNet-1k train dataset.

# K    ABLATION: CHANGING THE GUIDANCE WEIGHT

In this section, we test if the diversity improvements can be achieved by changing the classifier-free guidance weight. We find that it does improve diversity, however adding our SPELL on top consistently increases the performance further. We use the same SPELL hyperparameters as in the main paper for Latent Diffusion, namely $r = 20$ and overcompensation 1.6.

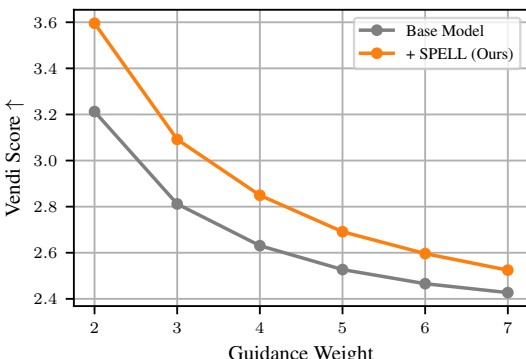

Figure 18: Our repellency added to the Latent Diffusion model with different classifier-free guidance weights.

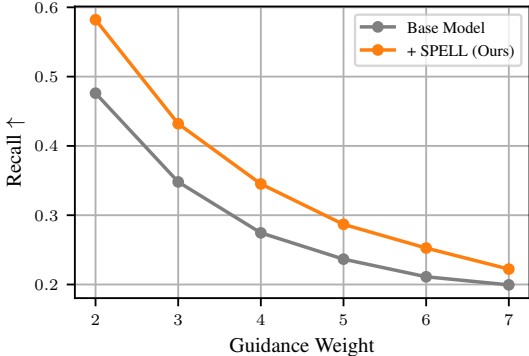

Figure 19: Our repellency added to the Latent Diffusion model with different classifier-free guidance weights.

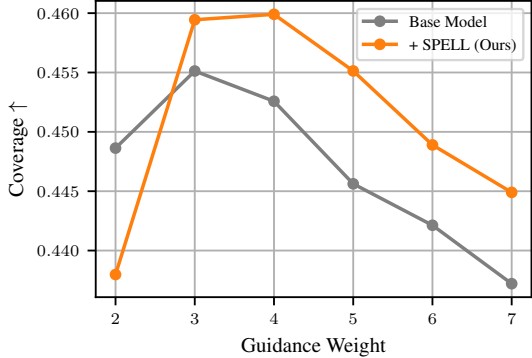

Figure 20: Our repellency added to the Latent Diffusion model with different classifier-free guidance weights.

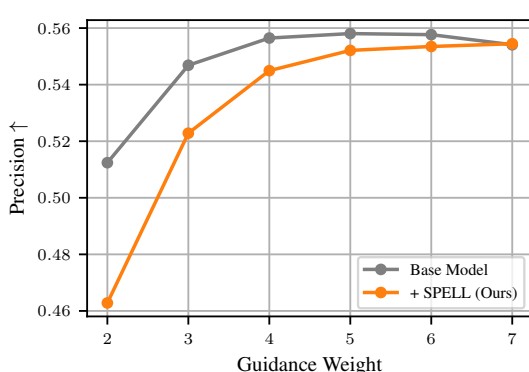

Figure 21: Our repellency added to the Latent Diffusion model with different classifier-free guidance weights.

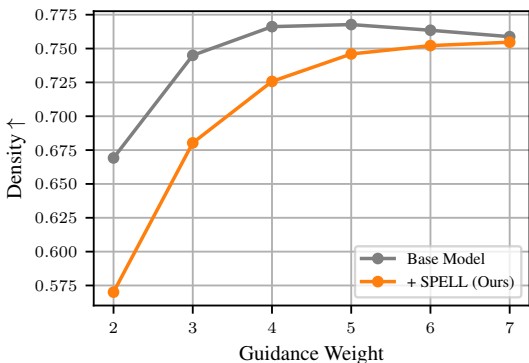

Figure 22: Our repellency added to the Latent Diffusion model with different classifier-free guidance weights.

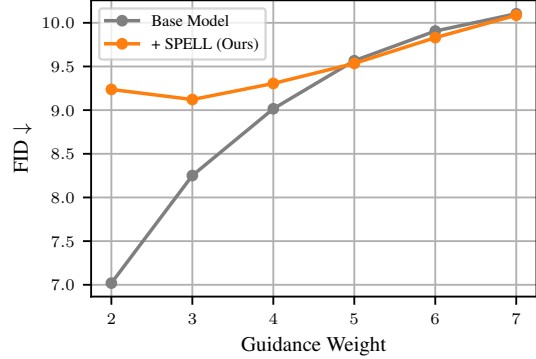

Figure 23: Our repellency added to the Latent Diffusion model with different classifier-free guidance weights.

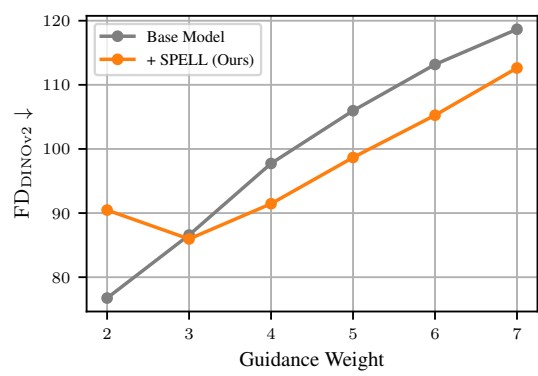

Figure 24: Our repellency added to the Latent Diffusion model with different classifier-free guidance weights.

## L  FURTHER DIVERSITY EXAMPLES

In order to extend Figure 1, we provide further examples of Simple Diffusion without and with SPELL in Figure 25 to Figure 34. The prompts are chosen from MS COCO, which Simple Diffusion was not trained on. As opposed to Figure 1, this features both of SPELL's capabilities: Intra-batch repellency (every row is a batch of size four), and inter-batch repellency from previous batches, which we treat as the shielded set. The examples affirm qualitatively that SPELL increases the diversity of generated images. Notably, this is without introducing visual artifacts and without lowering the prompt adherence, which other baselines like IG are prone to, see Table 4 and Figure 4.

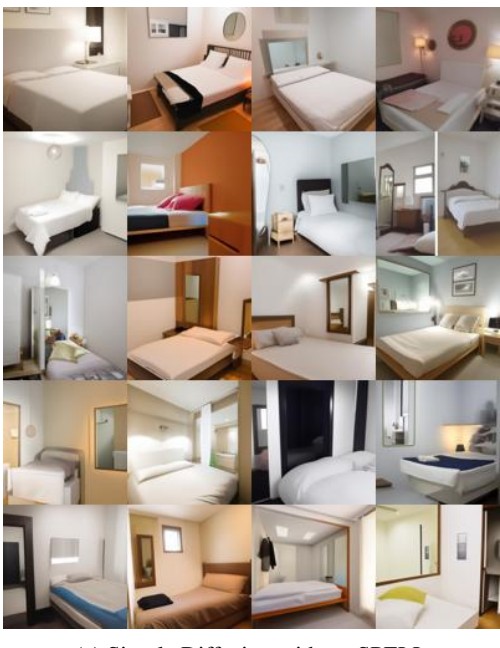 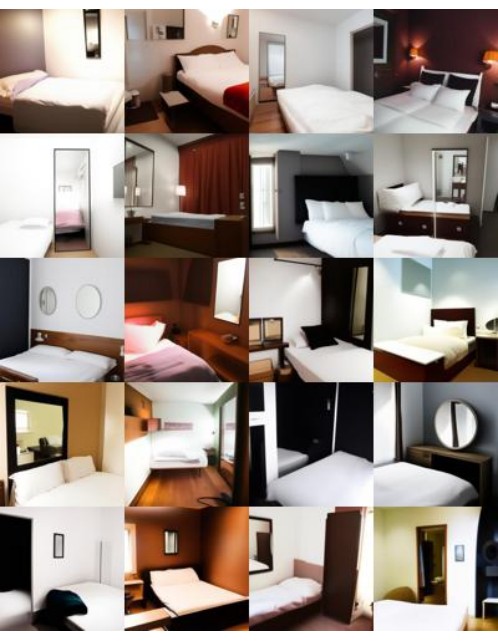

(a) Simple Diffusion without SPELL  (b) Simple Diffusion + SPELL

Figure 25: Images generated with Simple Diffusion without and with SPELL for the MS COCO prompt *"A bed and a mirror in a small room."*. Five batches (rows) with each four images, with both intra- and inter-batch repellency, with the same seeds as the runs without SPELL.

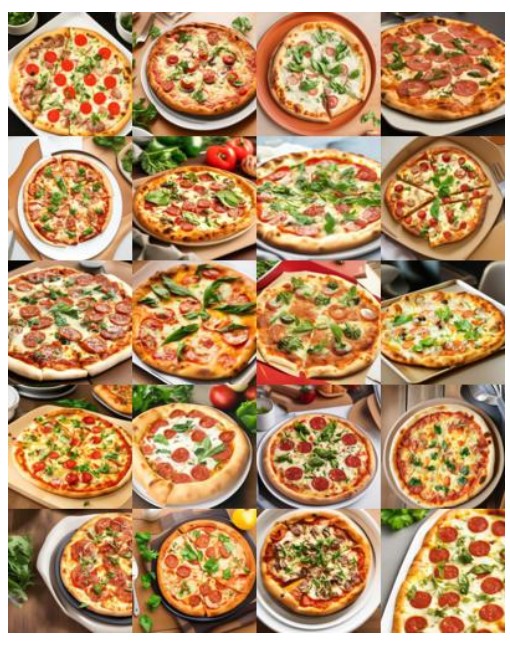 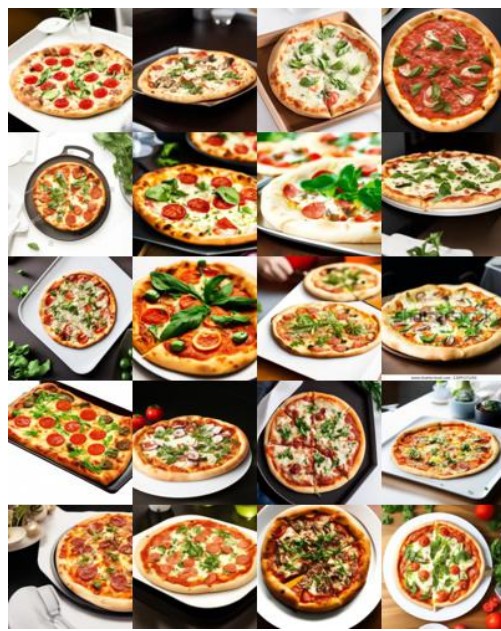

(a) Simple Diffusion without SPELL
(b) Simple Diffusion + SPELL

Figure 26: Images generated with Simple Diffusion without and with SPELL for the MS COCO prompt *"Baked pizza with herbs displayed on serving tray at table."*. Five batches (rows) with each four images, with both intra- and inter-batch repellency, with the same seeds as the runs without SPELL.

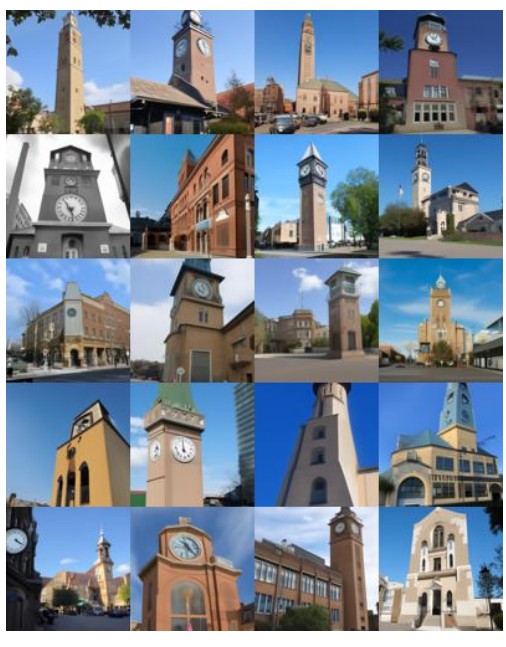 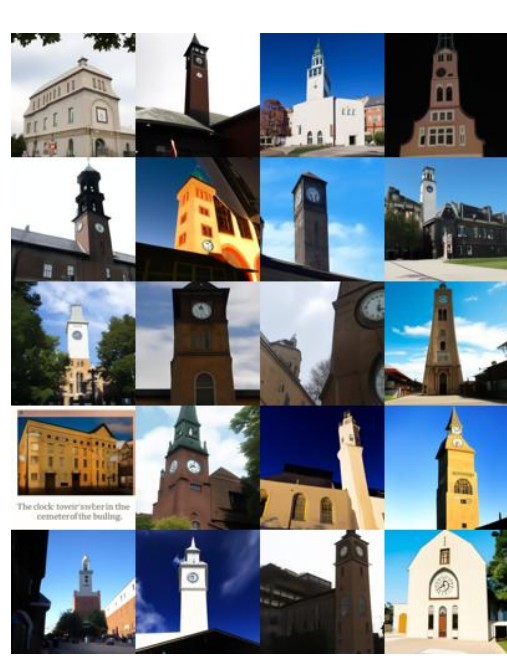

(a) Simple Diffusion without SPELL
(b) Simple Diffusion + SPELL

Figure 27: Images generated with Simple Diffusion without and with SPELL for the MS COCO prompt *"The clock tower is in the center of the building."*. Five batches (rows) with each four images, with both intra- and inter-batch repellency, with the same seeds as the runs without SPELL.

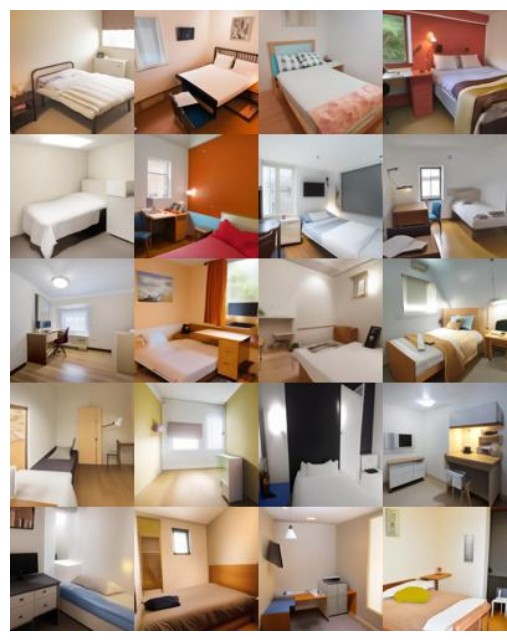 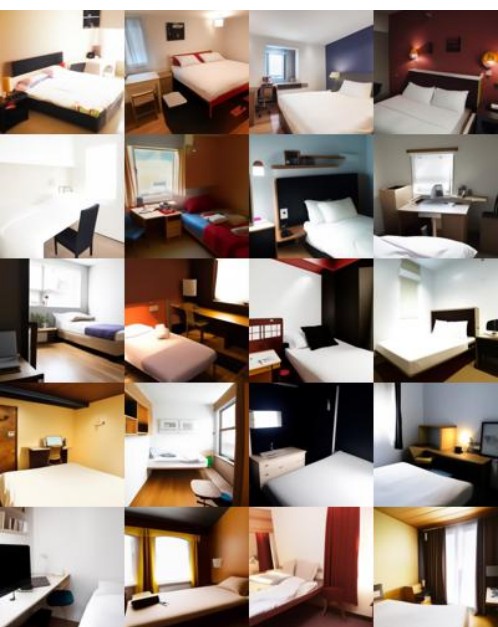

(a) Simple Diffusion without SPELL      (b) Simple Diffusion + SPELL

Figure 28: Images generated with Simple Diffusion without and with SPELL for the MS COCO prompt *"A bed and desk in a small room."*. Five batches (rows) with each four images, with both intra- and inter-batch repellency, with the same seeds as the runs without SPELL.

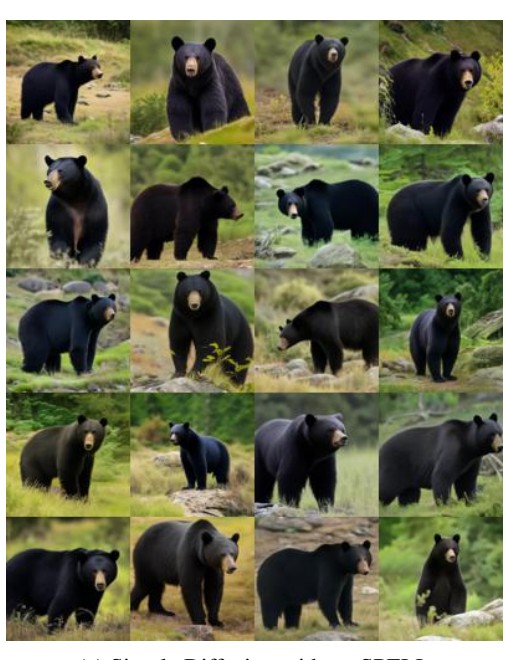 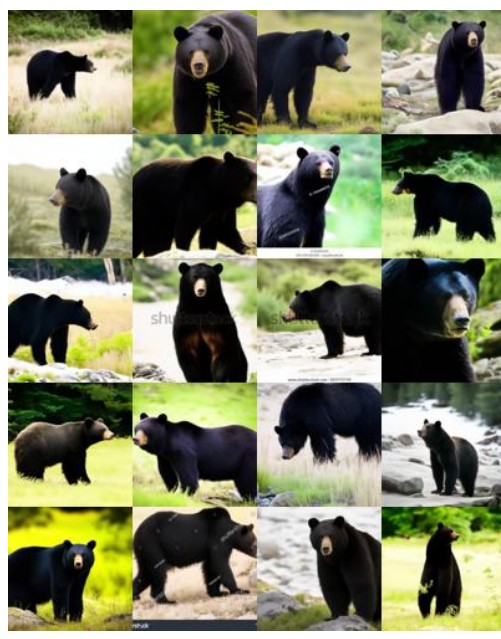

(a) Simple Diffusion without SPELL      (b) Simple Diffusion + SPELL

Figure 29: Images generated with Simple Diffusion without and with SPELL for the MS COCO prompt *"A furry, black bear standing in a rocky, weedy, area in the wild."*. Five batches (rows) with each four images, with both intra- and inter-batch repellency, with the same seeds as the runs without SPELL.

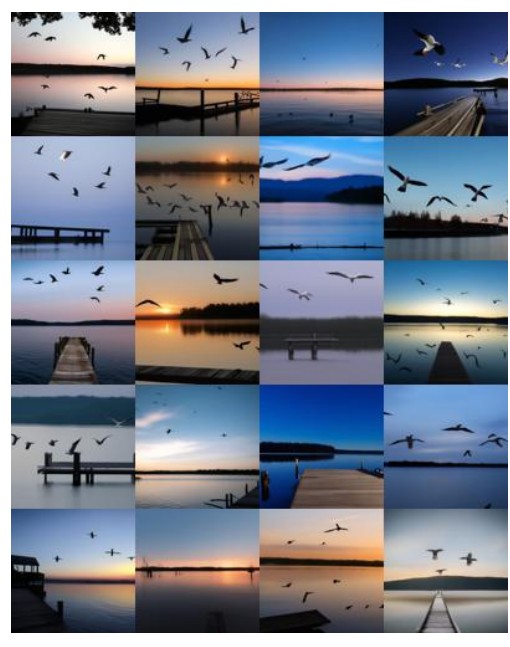 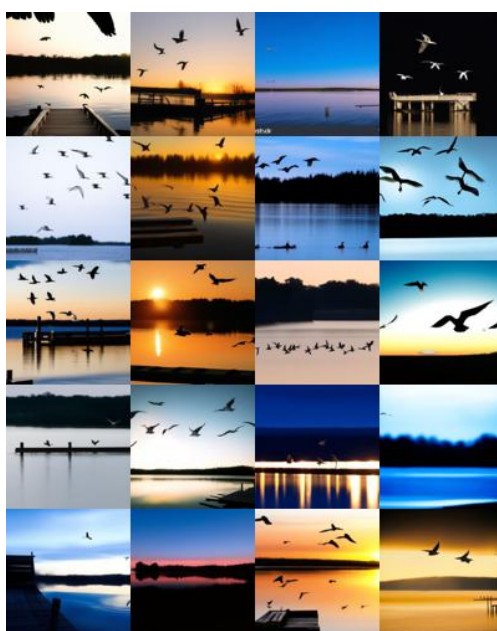

(a) Simple Diffusion without SPELL       (b) Simple Diffusion + SPELL

Figure 30: Images generated with Simple Diffusion without and with SPELL for the MS COCO prompt *"A group of seagulls are flying over a wooden dock that is sitting in a lake during the early part of the evening."*. Five batches (rows) with each four images, with both intra- and inter-batch repellency, with the same seeds as the runs without SPELL.

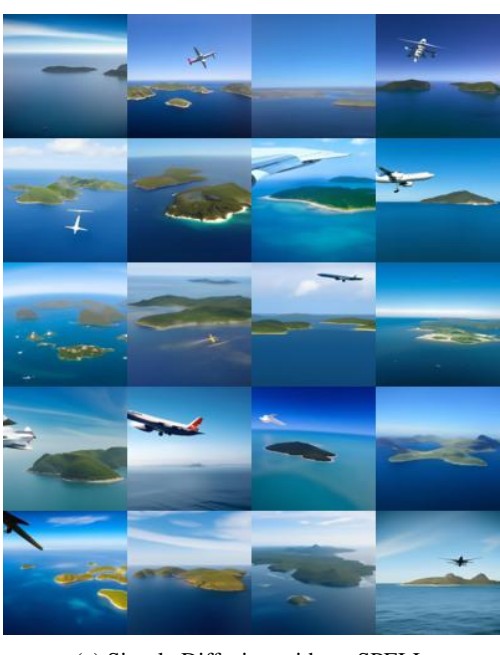

(a) Simple Diffusion without SPELL       (b) Simple Diffusion + SPELL

Figure 31: Images generated with Simple Diffusion without and with SPELL for the MS COCO prompt *"A plane flies over water with two islands nearby."*. Five batches (rows) with each four images, with both intra- and inter-batch repellency, with the same seeds as the runs without SPELL.

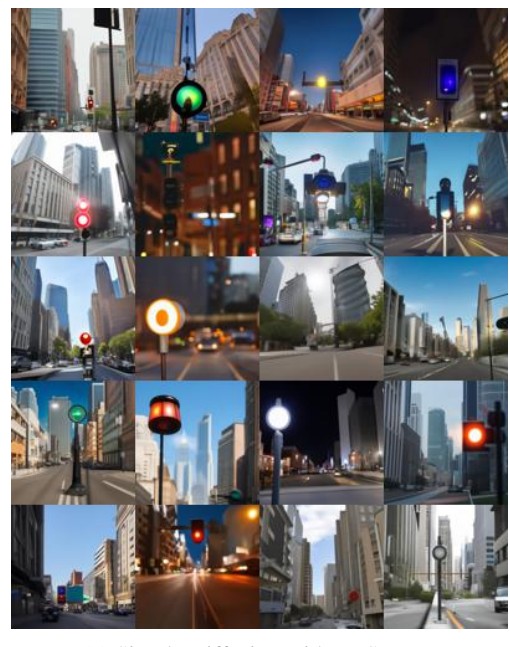 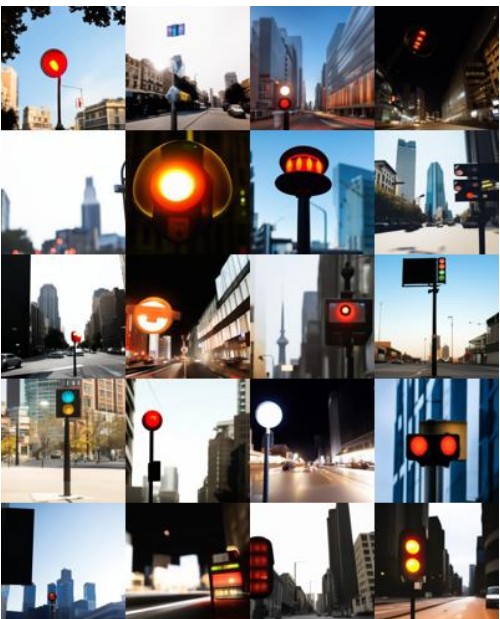

(a) Simple Diffusion without SPELL        (b) Simple Diffusion + SPELL

Figure 32: Images generated with Simple Diffusion without and with SPELL for the MS COCO prompt *"A traffic light over a street surrounded by tall buildings."*. Five batches (rows) with each four images, with both intra- and inter-batch repellency, with the same seeds as the runs without SPELL.

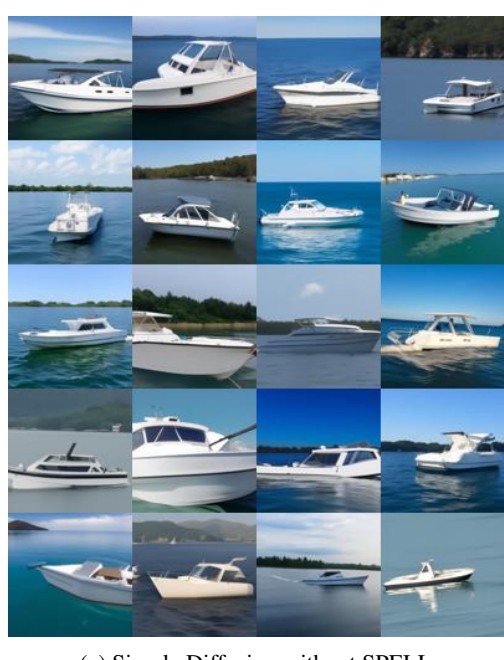 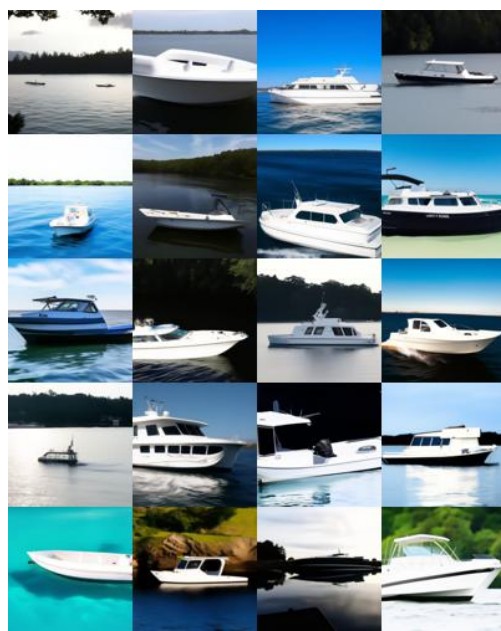

(a) Simple Diffusion without SPELL        (b) Simple Diffusion + SPELL

Figure 33: Images generated with Simple Diffusion without and with SPELL for the MS COCO prompt *"a white boat is out on the water"*. Five batches (rows) with each four images, with both intra- and inter-batch repellency, with the same seeds as the runs without SPELL.

1782
1783
1784
1785
1786
1787
1788
1789
1790
1791
1792
1793
1794
1795
1796
1797
1798
1799
1800
1801
1802
1803
1804
1805
1806
1807
1808
1809
1810
1811
1812
1813
1814
1815
1816
1817
1818
1819
1820
1821
1822
1823
1824
1825
1826
1827
1828
1829
1830
1831
1832
1833
1834
1835

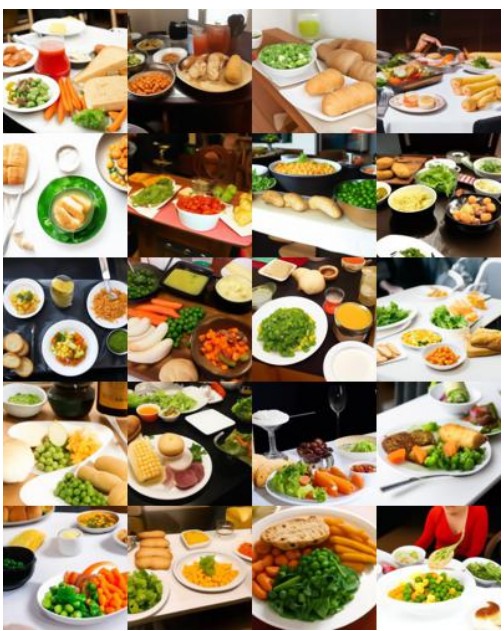

(a) Simple Diffusion without SPELL         (b) Simple Diffusion + SPELL

Figure 34: Images generated with Simple Diffusion without and with SPELL for the MS COCO prompt *"A table layed out with food such as, salad, steamed peas and carrots, steamed corn, and bread rolls."*. Five batches (rows) with each four images, with both intra- and inter-batch repellency, with the same seeds as the runs without SPELL.