# OpenReview forum: "Sparse Repellency for Shielded Generation in Text-to-Image Diffusion Models"
_ICLR.cc/2025/Conference — Submitted to ICLR 2025_

### Official Review · Reviewer_H25b · 2024-11-02

**Soundness:** 2
**Presentation:** 4
**Contribution:** 3
**Rating:** 6
**Confidence:** 4

**Summary:**

This paper introduces a novel post-training guidance mechanism, SPELL, which primarily addresses the training-set protection issue and the diversity problem of image diffusion models. SPELL is designed to repell the latents away from a trajectory that is close to a protected image set or from other latents within the same inference batch. It dynamically introduce small corrections to the latents in a way that is sparse and only triggered when the predicted trajectory is too closely to a reference domain. The authors evaluate SPELL on multiple state-of-the-art open-sourced diffsion models, showing its effectiveness. They also provide comparisons to other previous approaches that are also aimed at addressing diversity or with protected image set, which show some superior results on selected trading-off plots.

**Strengths:**

1. The overall problem that this paper addresses is one of the important issues that current diffusion-based image generation models possess, which adds to the value of motivations for this paper.
2. This paper is well-written and easy-to-understand. Notations within the background section and the method section are self-contained and clear to follow. Fig. 2 further adds readability.
3. The method this paper proposed is novel, which provides conceptual insights particularly in the method section (Sec. 4).
4. Experiments contain both ablation studies and comparisons to other methods. Fig. 3 show the effect of SPELL's only parameter **r**, in which we see effectiveness especially around 10-20.

**Weaknesses:**

MAJOR:
1. The core issue for this paper is the soundness in terms of the superior effectiveness compared to other similar methods. In Fig. 4, there is only comparisons on recall-precision, converage-density, and vendi-clip trade-off, while other concerning metrics in Tab. 1, such as $\text{FID}$, $\text{FD}_\text{DINOv2}$, are not included. I also failed to find reasoning on why only these three metric-pairs are selected.
2. This discussion of the fundamental methodology difference towards **Particle Guidance** on Page 5 is not convincing, as it seems SPELL can be treated as a special case of Particle Guidance when the energy potential $\phi_t$ is simply calculating the difference.
3. This paper provides abundant unconditional results with protected image set being ImageNet-1k in Fig. 17 in the appendix, but it seems that the only qualitative results for diversity is in Fig. 1. This paper could benefit from more concrete visual evidences.

MINOR:
1. In contributions, bullet point 3, **generated** is misspelled.
2. In contributions, bullet point 2, explanations on the **future looking** feature is quite not straight-forward to understand, I'd suggest keeping it brief here as a bullet point in contributions, and further explain it in method section with mathematical symbols, such as $x_t, x_0$.

**Questions:**

1. As it is difficult to show all possible trade-offs, a better way of giving concrete comparisons would be adding detailed tables for each of the methods. Each table shows all results with rows to be parameters, and columns to be all the metrics. Adding such tables would surely address my core concern, but due to limited time, it is also promising if the reasoning of choosing these trade-offs are persuasive and convincing.
2. I'm generally not quite sure if SPELL could be treated as a special case of Particle Guidance in terms of intra-batch diversity. A brief explanation would be sufficient.
3. This paper would also benefit from providing more diversity results, but it is understandable if this is infeasible considering the limited time for rebuttal.

---

> ### Author Response · Authors · 2024-11-15
> **Reviewer H25b: Rebuttal discussion**
>
> Many thanks for your encouraging comments and for praising our presentation with a score of 4. We are grateful for your many questions which we try to answer within the timeline of this rebuttal.
>
> > **The core issue for this paper is the soundness in terms of the superior effectiveness compared to other similar methods. In Fig. 4, there is only comparisons on recall-precision, converage-density, and vendi-clip trade-off, while other concerning metrics in Tab. 1, such as FID, FIDINOv2, are not included. I also failed to find reasoning on why only these three metric-pairs are selected.**
>
> This is a great point. The reason we have used these metric pairs is that they are popular diversity (y-axis)  vs. quality (x-axis) pairs. But following your request, we have added a fourth plot in Fig.2, with FD_DINOv2 vs CLIP Score. We have also added Table 4, with all metrics, to provide an exhaustive view of all metrics.
>
>
> > **This discussion of the fundamental methodology difference towards Particle Guidance on Page 5 is not convincing, as it seems SPELL can be treated as a special case of Particle Guidance when the energy potential ϕt  is simply calculating the difference.**
>
> Thanks for this comment. We agree that we were not clear enough in this section, starting with the title _(Intra-batch) SPELL as Particle Guidance_ which was, in retrospect, confusing. We have corrected this section in the paragraph you reference. Given more space we can expand a bit further the discussion. To be more detailed:
>
> - PG was proposed for intra-batch diversity following principles from the literature on interacting particles (_not_ to protect generation away from a reference set). We believe this viewpoint has guided two important choices: casting modifications of the trajectories as gradients of an interaction potential + use of a soft-decaying kernel that considers _exhaustively all_ interactions. This means that intra-batch similarity in PG is _always_ guiding / modifying the sampling of particles, throughout time and w.r.t. samples.
>
> - By contrast, SPELL is not defined as an energy minimizing principle, but follows instead from geometric principles (Fig. 1) which cannot, to our current knowledge, fit into such a “gradient” based perspective (we tried but experimental evidence suggests that SPELL interventions in Eq. 8 are not conservative). By doing away with this “interaction energy” principle we lose PG’s mathematical interpretation, but we gain efficiency and the ability to simply state our goal of making sure the trajectories almost always flow normally, and are only “bumped” when strictly needed (both w.r.t. batch but also in time, see Fig. 5(b), Fig. 9~16). In our view, that sparsity w.r.t samples _and_ time is crucial to scale in our application to image protection, but also to get unperturbed trajectories, and our experiments validate this intuition.
>
> > **This paper could benefit from more concrete visual evidences.**
>
> This is a great suggestion. We are getting the computational resources to generate further example images, and will come back to you very soon.
>
> > **As it is difficult to show all possible trade-offs, a better way of giving concrete comparisons would be adding detailed tables for each of the methods. Each table shows all results with rows to be parameters, and columns to be all the metrics. Adding such tables would surely address my core concern, but due to limited time, it is also promising if the reasoning of choosing these trade-offs are persuasive and convincing.**
>
> Thanks for the suggestion! We have added Table 4. We hope that this addresses your core concern.
>
> > **I'm generally not quite sure if SPELL could be treated as a special case of Particle Guidance in terms of intra-batch diversity. A brief explanation would be sufficient.**
>
> We hope our answer above assuages your concerns.
>
> To recapitulate, the two fundamental differences with PG lie in (i) adding guidance terms that are _not_ grad-potentials (ii) designing specifically very sparse repellence terms, with sparsity in two senses:
> - They act on trajectories rarely over time, and most interventions happen early and then vanish, see Fig. 5b. This is thanks to our focus, from the start, on _expected_ generation and not location at time $x_t$.
> - They only add sparse terms, to the extent that most of them are 0, when comparing to points in the reference set (either self-reference in intra-batch, or external reference for protection). This makes our method scale to 1.2M points as a reference set and leaves dynamics unperturbed *when perturbations are not needed*.
>
> > **This paper would also benefit from providing more diversity results, but it is understandable if this is infeasible considering the limited time for rebuttal.**
>
> This is a great point. As mentioned above, we are getting access to resources needed to generate these examples. We will get back to you soon.
>
> > **MINOR**
>
> Thanks for spotting these! We addressed these points in the revision.

---

> > ### Author Response · Authors · 2024-11-18
> >
> > We are happy to come back to you with new results that we have added to the revised paper, following your suggestion.
> >
> > > **This paper could benefit from more concrete visual evidences.** / **This paper would also benefit from providing more diversity results, but it is understandable if this is infeasible considering the limited time for rebuttal.**
> >
> > **We have now finished generating 400 further example images. We provide these examples of Simple Diffusion with and without SPELL in Figures 25 to Figure 34 in the revised paper.** The 10 prompts are chosen from MS COCO, which Simple Diffusion was not trained on. The examples affirm qualitatively that SPELL increases the diversity of generated images. Notably, this is without introducing visual artifacts and without lowering the prompt adherence, which other baselines like IG are prone to, see Table 4 and Figure 4.

---

> > > ### Comment · Reviewer_H25b · 2024-11-24
> > >
> > > Thank you so much for your timely response and your additional results. After carefully reviewing these new results and comments, I'm summarizing my previously mentioned three major concerns.
> > > - All the trade-off experiments provided good overall results compared to other methods. These results in Fig. 4 and Tab. 4 are not perfect, but good enough to convence me on this point.
> > > - The reasoning that SPELL is NOT a special case of PG well-convinced me. SPELL is indeed a sparse method that is only triggered on certain conditions, while PG is always adding a gradient term that manipulates the sampling process.
> > > - New qualitative results from Fig. 25 to Fig. 34 are generally good. Although the differences with and without SPELL on some cases are not obvious or straightforward, the overall performances would be sufficient.
> > >
> > > That said, I still have a minor question just out of curiosity. In Fig. 29(b), there are watermarks saying "ShutterStock" in some of the generated samples, which clearly is resulting from the WebVid dataset. Since the samples without SPELL applied are not producing such watermarks, I'm therefore wondering does it have anything to do with SPELL?

---

> > > > ### Author Response · Authors · 2024-11-26
> > > >
> > > > Thank you for taking the time for such an exceptional in-depth discussion! We’ve generated some more data to follow up on your intuition regarding shutterstock overlays.
> > > >
> > > > We’ve generated 1600 examples for Simple Diffusion without SPELL and 1600 with SPELL. Without SPELL, **62/1600** images have a shutterstock (or similar) overlay, with SPELL it’s **105/1600**.
> > > >
> > > > To confirm that this is a stable trend, we’ve also generated images with Latent Diffusion (which was trained on the same dataset as Simple Diffusion), where it’s **79/800** without SPELL and **98/800** with SPELL. While the second result could still be a random chance (Chi-Square independence test with Yates' continuity correction gives p-value = 0.15), the first result is beyond random (p=0.001), and also the effect size is quite measurable (7% vs 4% overlay rate).
> > > >
> > > > Two observations:
> > > > - We have noted that the copyright overlays tend to happen clustered at specific prompts. E.g., one motorcycle prompt has 21/32 images with overlay, and many other prompts have 0. So the distribution is quite skewed and seems to depend on the prompt.
> > > > - We have calculated pairwise distances within generated batches and found that the watermarks can serve to push away images from similar ones without the watermark and to pull together images with the same watermark.
> > > >
> > > > In our understanding the copyright overlays might serve as a “highway” between modes, which could  allow SPELL to easily explore new modes. SPELL only uses this if the training images / mode distribution for a given prompt actually includes these overlays.
> > > >
> > > > We can include these findings in the revision. We also believe that it is a good inspiration for future works: SPELL could be applied to only change parts of an image by simply masking the distance calculation. This could for example allow to explicitly suppress copyright overlays if one wishes so, or to remove a part of an image, shield its original content, and guarantee to generate truly new fill-ins with applications in anonymization. However, this would require changing the evaluation protocol severely, so we will denote it as an avenue for future research in the paper in the revision.

---

> > > > > ### Comment · Reviewer_H25b · 2024-11-26
> > > > > **Score Adjustments**
> > > > >
> > > > > Thank you for providing a thorough analysis in such a short time. This understanding and the extended variant to apply SPELL on partial images with masks are quite interesting to me.
> > > > >
> > > > > After considering the addressed concerns and the opinions from other reviewers, I am now **very confident to recommend accepting this paper (somewhere around a score of 7), while staying conservative and uncertain on higher scores**. The reasonings are as follows:
> > > > > - My recommendation on acceptance is drawn from the well-written paper, the sufficient results on experiments, and particularly, the interesting concept and approach of sparsely penalizing the sampling process. The idea itself, regardless of how the experiments perform, is worth publishing.
> > > > > - The reason why I'm not sure on recommending higher scores is that the experiments, both qualitative and quantitative, still have certain imperfections, and I'd refer to other reviewers on this point. That said, I am extremely grateful to the authors for providing these massive new results in such a short notice.

---

> > > > > > ### Author Response · Authors · 2024-11-27
> > > > > >
> > > > > > Thank you again for engaging with us in this rebuttal, we are grateful for your time and your many insights that have helped us improve the submission.
> > > > > >
> > > > > > > **I am now very confident to recommend accepting this paper (somewhere around a score of 7), while staying conservative and uncertain on higher scores.**
> > > > > >
> > > > > > We are grateful for your acceptance recommendation.
> > > > > >
> > > > > > On the point of selecting a score that properly reflects your recommendation, we believe this is of course, ultimately, your decision and yours only.
> > > > > >
> > > > > > Since the current scale at ICLR does not include a numerical rating of 7 nor 9, the alternatives are
> > > > > > * _6: Marginally above the acceptance threshold_
> > > > > > * _8: Accept, good paper_
> > > > > > * _10: Strong accept, should be highlighted at the conference_
> > > > > >
> > > > > > With these text labels attached to the numerical scores in mind, we do not believe that an 8 in ICLR should be treated in this scale as a “high score” as in other conferences (in ICML/NeurIPS, it would be _“8: Strong Accept”_). It just means a recommendation for acceptance, expressing confidence that you believe the paper will be an interesting addition to the program (as a poster or else), so the ICLR _“8: Accept”_ is more similar to a 7 in other venues (ICML/Neurips defines _“7: Accept”_). A 6, on the other hand, means that you are more undecided about its fate, erring on the side of caution.
> > > > > >
> > > > > > > **The idea itself, regardless of how the experiments perform, is worth publishing.**
> > > > > >
> > > > > > Many thanks for this assessment.
> > > > > >
> > > > > > > **the experiments, both qualitative and quantitative, still have certain imperfections, and I'd refer to other reviewers on this point.**
> > > > > >
> > > > > > If you have other specific concerns or questions on our experiments, we will do our best to answer them. We remain at your disposal to clarify them and welcome this opportunity to further improve the paper. At this point, we believe that we have answered all outstanding questions, as summarised in our general answer.

---

### Official Review · Reviewer_94st · 2024-11-04

**Soundness:** 3
**Presentation:** 4
**Contribution:** 3
**Rating:** 5
**Confidence:** 4

**Summary:**

This paper proposes a novel way to diversify diffusion generations by introducing repellency terms to the diffusion SDE. It achieves the diversity of generated images from one prompt and/or prevention of similar generation to the reference set, which is one of the significant challenges for real users.

**Strengths:**

- This paper tackles a very practical problem of repetitive generations of text-to-image diffusion models to both protective images in the training set and previously generated images.
- The proposed method, repellency terms, has a concrete background and intuitive to solve the problem.
- Thorough empirical investigations provide enough understanding of how SPELL can increase the diversity of generated images, and the advantages for real-world applications.

**Weaknesses:**

- While SPELL can be used for any diffusion pipelines, the effectiveness of SPELL for smaller models or domains other than ImageNet is not fully investigated.
- The efficiency of SPELL is validated for ImageNet class or simple text prompts where the diversity within a text prompt is huge. Evaluation for more complex text prompts that align with more practical usage of text-to-image diffusion models would validate the effects of SPELL more.
- As noted by authors, the proposed SPELL does not provide a very tight guarantee to avoid generations of similar images to the reference set, which can limit the applicability of SPELL for high-risk cases.

**Questions:**

- Applying repellency terms based on the current state x_t instead of the expected final output seems applicable for the intra-batch repellency case, providing better diversity to a batch of generated images. Can it be one of the baselines to compare SPELL for the intra-batch case?
- It seems like SPELL forces each trajectory to arrive near the boundary of other balls (shields). Can further methods (something similar to momentum or just larger overcompensation) improve the diversity of generated images?

---

> ### Author Response · Authors · 2024-11-15
> **Reviewer 94st: Rebuttal discussion**
>
> We are very grateful for your detailed review, your encouragements and your score of *4* for presentation. Here are a few answers to your concerns:
>
> > **While SPELL can be used for any diffusion pipelines, the effectiveness of SPELL for smaller models or domains other than ImageNet is not fully investigated.**
>
> It was very tempting for us to use text2image as a testing ground for SPELL because this facilitates visual inspection of results and is well suited to the ICLR audience. Additionally, since one of our main claims is that our interventions are sparse (unlike PG) and can scale, considering very large reference sets (e.g. 1.2M images in ImageNet) was crucial. We believe that using our code will run into less challenges on smaller scale problems and other fields.
>
> > **SPELL does not provide a very tight guarantee to avoid generations of similar images to the reference set, which can limit the applicability of SPELL for high-risk cases.**
>
> Thanks for this great point. SPELL is already the first method that is close to a perfect guarantee due to its geometric intervention rules (Eq. 5). In the protection experiment in S.4.6, we previously showed a 99.45% protection rate for EDMv2 + SPELL, compared to 92.40% for the bare EDMv2.
>
> In fact, we can introduce a tradeoff, where the fast-NN search is more accurate (and longer) by increasing the number of searched Voronoi cells. We can reach a 99.84% success rate with a more accurate search (see updated Table 2 in revision). We have not optimized yet the parameters of the fast-NN search (currently this is run, sub-optimally, on CPU). We believe these overheads can be significantly reduced.
>
> | Model     	| Searched cells | Time per image (s) ↓ | Generated images far enough from all ImageNet images ↓ |
> |---------------|----------------|-----------------------|---------------------------------------------------------|
> | EDMv2     	| -          	| 2.434             	| 92.40%                                          	|
> | + SPELL   	| 1          	| 4.633             	| 98.92%                                          	|
> | + SPELL   	| 2          	| 6.057             	| 99.45%                                          	|
> | + SPELL   	| 3          	| 7.790             	| 99.67%                                          	|
> | + SPELL   	| 5          	| 9.949             	| 99.78%                                          	|
> | + SPELL   	| 10         	| 13.545            	| 99.84%                                          	|
>
>
> > **Applying repellency terms based on the current state x_t instead of the expected final output seems applicable for the intra-batch repellency case, providing better diversity to a batch of generated images. Can it be one of the baselines to compare SPELL for the intra-batch case?**
>
> Thanks for this great comment.
>
> When the repellency terms only depend on the current state $x_t$ (and when the repellency terms are the grad of an interaction potential, and we restrict to intra-batch repellency) what you suggest is exactly particle guidance ([Corso et al. 24]), as detailed in lines 250~265. PG is featured prominently in our experiments (it's one of the baselines in Fig. 4, and added to Table 4 in the revised paper).
>
> Note however that PG interactions are not sparse, and cannot therefore be realistically extended to our large scale protection experiment.
>
> > **It seems like SPELL forces each trajectory to arrive near the boundary of other balls (shields).**
>
> We will clarify this, but this is not really the case.
>
> What you describe would be somewhat our worst-case scenario, in which SPELL would fulfill the shielding requirement only when applied _at the last timestep_.
>
> Instead, as depicted in Fig. 1, when SPELL detects that a generation at time $t$ is _pointing_ towards a shield, it alters the diffusion direction to make it point outside of that shield. Ideally, these modifications happen as early as possible so that the diffusion can explore a different mode instead and perturb as little as possible the later (image-detail defining) stages of the diffusion, to avoid visual artifacts (Fig. 17).
>
> Luckily, following our intuition, SPELL interventions occur mostly in early diffusion timesteps, as shown in the density plot in Fig. 5a, that shows many more interventions early on in diffusion at time $t=1$. This is tightly connected to the radius that is chosen (Fig. 5b) and is indeed confirmed when $r=20$. This is also extensively documents in Figures 9~16.
>
> > **Can further methods (something similar to momentum or just larger overcompensation) improve the diversity of generated images?**
>
> We have not tried momentum, but this is a great suggestion. A small overcompensation of 1.6 seems to work very well. One combination we explored is to both use a lower CFG weight to increase diversity and add SPELL. In Appendix I, Figures 18 - 24, we find that even when we already increase diversity by decreasing the CFG weight, adding SPELL still boosts further diversity.

---

> > ### Comment · Reviewer_94st · 2024-11-27
> >
> > Thank you for the detailed response about my review, and sorry for the late reply.
> >
> > I have one remaining question regarding the first question. If I understand correctly, the major difference between the proposed SPELL and PG is where the distance between two points is computed (the space of the expected final output for SPELL and the space of the current value for PG). Does the sparsity property of SPELL come from this distinction or some differences in the detailed mechanism, such as one used to find neighbors to repel?

---

> ### Author Response · Authors · 2024-11-27
>
> > **Thank you for the detailed response about my review, and sorry for the late reply.**
>
> Many thanks for taking the time to read our rebuttal. Your reply is not late at all, since the deadline for discussion has now been extended.
>
> > **I have one remaining question regarding the first question. If I understand correctly, the **major difference** between the proposed SPELL and PG is where the distance between two points is computed (the space of the expected final output for SPELL and the space of the current value for PG).**  Does the sparsity property of SPELL **come from this distinction** or some differences in the detailed mechanism, **such as one used to find neighbors to repel?**
>
> Thanks for these questions. SPELL and PG are indeed closely related as we highlight in the paper L.246.
>
> To go back to their definition, the formulation of PG (Eq. 4 in their paper, L.249 in ours) defines an interaction potential (using RBF kernels) on particle positions $x_t$. The interventions that result from that potential are dense (the particle’s trajectories are always updated, even when they are far away, regardless of being neighbors or not) and exhaustive (all $B^2$ kernel values for a batch of size $B$ impact the trajectory).
>
> SPELL moves away from that model in two orthogonal aspects: As you point out, SPELL (1) is defined as “future looking”, in that it considers the expected generation of these points to guide trajectory rather than $x_t$ ; and (2) SPELL uses sparse updates, by definition, in the way we set these interventions in **Eq. (5)**. The notion of neighbours kicks in because of sparsity.
>
> Of these two, **the second factor, sparsity of interventions is the major difference, not the “future looking” aspect**, as already clarified in **L.261** of the revision. While PG could be used in principle with the same “future looking” approach, the crucial idea in PG is to posit that particles should be constantly re-updated, using a smooth kernel, as is common in the interacting particles literature they refer to (their Section 4). This is different from SPELL.
>
> Our decision to use a sparse intervention model is highlighted in the title of our paper and the name of our method. This is the major difference, which leads to:
> - applicability to large scale protection scenarios, where reference sets are of the order of millions (When generating a batch `B`, PG would need a costly computation of `B x 1M` Gaussian kernel values at each diffusion step if it were adapted to that task, and those 1M contributions would likely have unintended consequences for quality of generation)
> - qualitative improvements, because sparse interventions result intuitively in far fewer changes in the particle trajectories. **This is our analysis in Fig.5 + 9~16**, as well as results in Fig. 4.
>
> Hence, as a **TL;DR** summary, SParsity in SPELL is the major differentiator w.r.t. PG. Sparsity comes from geometric considerations (sketched in Fig. 2), and from the definition of our repulsive terms (our choice in Eq. 5). Sparsity does not come from the additional consideration of using “future looking” $E[X_0|X_t=x_t]$ instead of $x_t$.

---

> > ### Comment · Reviewer_94st · 2024-11-30
> >
> > Thank you for your respectful response. I asked since I was confused by the previous answer for my Q1.
> >
> > Now I understand the major contribution of SPELL and its differences from PG.
> >
> > I appreciate your help.

---

### Official Review · Reviewer_V3e4 · 2024-11-04

**Soundness:** 3
**Presentation:** 3
**Contribution:** 3
**Rating:** 6
**Confidence:** 1

**Summary:**

This paper introduces a novel guidance mechanism called SPELL (Sparse Repellency) aimed at enhancing diversity and protecting certain reference images during the generation process in text-to-image diffusion models. This approach addresses two common challenges with diffusion models: the tendency to produce repetitive images for the same prompt and the potential risk of inadvertently recreating training images, which raises privacy and copyright concerns. In summary, SPELL is a post-training intervention that enhances image diversity and safeguards specific images by selectively adjusting generation paths. This method offers a practical solution for more diverse and privacy-respecting image generation in diffusion models.

**Strengths:**

1. The paper is well written.
2. The addressed problem is important in diffusion models.

**Weaknesses:**

I am not familiar with this field, but I find the issue addressed in this paper to be interesting and important. AC can disregard my opinion and score.

**Questions:**

N/A

---

> ### Author Response · Authors · 2024-11-14
> **Reviewer V3e4: Rebuttal discussion**
>
> Many thanks for reading our paper despite this not being your area of expertise. Still, we are very happy to see that despite this mismatch you had a positive impression of our paper overall. ICLR papers should be tailored to reach a wide readership, and we took great efforts to make our paper easy to parse, through e.g. Fig. 1 and other illustrations.
>
> > **I find the issue addressed in this paper to be interesting and important.**
>
> Indeed, we believe these are core issues that appear naturally when letting end-users interact with diffusion models.
>
> In our first application (intra-batch diversity), we were happy to advance the recent SOTA set in ICLR 2024  (https://arxiv.org/abs/2310.17347 and https://arxiv.org/abs/2310.13102) and NeurIPS 2024 (https://arxiv.org/abs/2404.07724).
>
> As for our second application (protecting images), we are the first (to our knowledge) to propose a method for this task that works at such a scale, with an impressive reference set of more than 1 million images.

---

> > ### Comment · Reviewer_V3e4 · 2024-11-27
> >
> > Thank you for your reply!

---

### Official Review · Reviewer_EnjK · 2024-11-05

**Soundness:** 2
**Presentation:** 3
**Contribution:** 2
**Rating:** 6
**Confidence:** 4

**Summary:**

This paper presents a novel technique called SPELL to address the challenge of shielded generation and improve generation diversity in text-to-image diffusion models. SPELL shields the model from replicating protected images and promotes intra-batch diversity by adding sparse repellency terms to the diffusion process, guiding generated images away from a reference set and the images in the same batch. The authors demonstrate the effectiveness of their methods via experiments on the state-of-the-art diffusion models. Compared to the previous works, SPELL achieves the best trade-off between image diversity and generation quality. Further, the authors empirically show that SPELL is scalable with a large reference set.

**Strengths:**

- The paper tackles a timely and practically-relevant problem supported by a fair amount of experiments. Shielded generation of text-to-image diffusion models is an area with limited prior research, making this work particularly valuable.
- Overall, the paper is clearly written and easy to follow.

**Weaknesses:**

- One key weakness of this paper is the lack of experiments regarding the main trade-offs against baseline methods. For example, while Table 1 indicates that SPELL has a minor trade-off in precision, the authors do not compare this trade-off with the baselines. Figure 4 is the only comparative result provided, but it lacks an analysis of image quality. Specifically, I would like to know if all models in Figure 4 are capable of similar  generation quality, say in terms of FID.
- Since SPELL is a training-free sampling method, the authors should also provide a quantitative analysis regarding inference time. For instance, an analysis of average wall-clock time compared with baseline methods, or testing with larger reference dataset sizes, would be helpful for readers.
- In the main qualitative analysis in Section 4.5, Figure 6 does not convincingly illustrate improved image generation diversity. For example, the fourth image is repelled from the third image, and it's unclear why this image is closer to the third image than to the first or second. Similarly, in the 13th image, the only notable difference is the color of the ball, and yet the blue ball is the most common color in prior images. Additionally, it would be beneficial if the authors provided examples with multiple image batches, other than a single-image batch.
- Regarding Figure 7, I wonder whether the $L_2$ distance-wise nearest neighbor search was the best choice. This is because many images in the third row (EDM + SPELL) seem more similar to the second row (ImageNet neighbor for EDM) rather than the fourth row (ImageNet neighbor for EDM+SPELL).

**Questions:**

- In line 359, it states that "precision and density decrease slightly in 5 out of 6 models." However, according to Table 1, isn't this actually the case for 4 out of 6 models? Please correct me if I am wrong.

---

> ### Author Response · Authors · 2024-11-14
> **Reviewer EnjK: Rebuttal discussion (part 1)**
>
> We would like to thank Reviewer **EnjK** for their encouraging comments and constructive review. We did our best within the time constraints to address all of the points that you have raised, and will do our best to answer other concerns.
>
> > **For example, while Table 1 indicates that SPELL has a minor trade-off in precision, the authors do not compare this trade-off with the baselines. Figure 4 is the only comparative result provided, but it lacks an analysis of image quality. Specifically, I would like to know if all models in Figure 4 are capable of similar generation quality, say in terms of FID.**
>
> This is a great point, thanks for raising it. Our goal (L. 403) was indeed to contrast diversity vs *quality* in 3 different plots, 3 different Pareto fronts, using Precision / Density / Clip scores. Following your request, **we have added a fourth plot in Fig. 4, with FD_DINOv2 vs CLIP Score**, as can be seen in our updated draft. We can add other similar figures if you think they would be relevant.
>
> **We have also added the following table (Table 4, p.21 in draft)**, with all metrics, to provide readers with an exhaustive view.
>
> |Method|Recall|Vendi Score|Coverage|Precision|Density|FID|$FD_\text{DINOv2}$|CLIP Score|
> |----------------------------------------------------|--------|-------------|----------|-----------|---------|-------|-----------|--------|
> |Base Model|0.237|2.527|0.446|0.558|0.768|9.566|105.967|27.789|
> |Particle Guidance, strength=1024|0.099|1.987|0.249|0.300|0.326|84.115|705.661|24.470|
> |Particle Guidance, strength=512|0.230|2.753|0.378|0.443|0.534|23.106|286.093|26.740|
> |Particle Guidance, strength=256|0.252|2.656|0.429|0.523|0.682|11.934|154.897|27.440|
> |Particle Guidance, strength=128|0.248|2.591|0.447|0.553|0.754|9.442|109.257|27.704|
> |Particle Guidance, strength=64|0.245|2.561|0.449|0.559|0.771|9.072|101.796|27.781|
> |Particle Guidance, strength=32|0.235|2.528|0.445|0.557|0.763|9.724|108.382|27.812|
> |Particle Guidance, strength=16|0.236|2.529|0.446|0.557|0.764|9.596|107.041|27.813|
> |Interval Guidance, [0.1,0.9]|0.372|2.840|0.455|0.537|0.730|8.385|85.871|27.453|
> |Interval Guidance, [0.2,0.9]|0.419|2.994|0.442|0.514|0.689|8.359|85.094|26.813|
> |Interval Guidance, [0.1,0.8]|0.470|3.174|0.448|0.500|0.663|7.507|76.104|27.215|
> |Interval Guidance, [0.3,0.9]|0.471|3.208|0.421|0.483|0.635|8.406|87.971|25.885|
> |Interval Guidance, [0.2,0.8]|0.518|3.340|0.434|0.478|0.624|7.478|75.250|26.544|
> |Interval Guidance, [0.1,0.7]|0.567|3.576|0.432|0.451|0.577|6.804|72.092|26.784|
> |Interval Guidance, [0.4,0.9]|0.525|3.495|0.395|0.442|0.569|8.623|96.611|24.630|
> |Interval Guidance, [0.3,0.8]|0.571|3.575|0.411|0.446|0.570|7.556|78.887|25.549|
> |Interval Guidance, [0.2,0.7]|0.614|3.770|0.417|0.426|0.536|6.771|72.972|25.979|
> |Interval Guidance, [0.1,0.6]|0.673|4.138|0.396|0.385|0.466|6.885|81.643|26.020|
> |CADS, mixture factor=0, tau_1=0.6|0.262|2.598|0.447|0.553|0.753|9.248|105.006|27.746|
> |CADS, mixture factor=0, tau_1=0.7|0.253|2.579|0.448|0.555|0.757|9.288|105.549|27.757|
> |CADS, mixture factor=0, tau_1=0.8|0.245|2.561|0.449|0.557|0.762|9.356|105.856|27.771|
> |CADS, mixture factor=0, tau_1=0.9|0.239|2.545|0.450|0.559|0.767|9.452|106.455|27.790|
> |CADS, mixture factor=0.001, tau_1=0.6|0.325|2.816|0.442|0.531|0.696|8.897|105.081|27.534|
> |CADS, mixture factor=0.001, tau_1=0.7|0.297|2.734|0.446|0.540|0.719|8.963|104.006|27.617|
> |CADS, mixture factor=0.001, tau_1=0.8|0.277|2.660|0.447|0.548|0.739|9.098|103.766|27.697|
> |CADS, mixture factor=0.001, tau_1=0.9|0.256|2.588|0.448|0.554|0.755|9.273|105.268|27.754|
> |CADS, mixture factor=0.002, tau_1=0.6|0.425|3.208|0.417|0.472|0.584|9.870|129.159|26.920|
> |CADS, mixture factor=0.002, tau_1=0.7|0.380|3.028|0.429|0.501|0.637|9.143|114.333|27.242|
> |CADS, mixture factor=0.002, tau_1=0.8|0.330|2.837|0.442|0.529|0.692|8.893|105.511|27.506|
> |CADS, mixture factor=0.002, tau_1=0.9|0.277|2.660|0.446|0.548|0.739|9.098|103.762|27.696|
> |SPELL, shield radius=40|0.370|2.998|0.437|0.500|0.631|13.072|140.841|27.397|
> |SPELL, shield radius=35|0.359|2.935|0.445|0.518|0.665|11.452|120.346|27.556|
> |SPELL, shield radius=30|0.337|2.856|0.451|0.531|0.695|10.349|106.753|27.655|
> |SPELL, shield radius=25|0.312|2.774|0.454|0.542|0.723|9.794|100.123|27.739|
> |SPELL, shield radius=20|0.287|2.691|0.455|0.552|0.746|9.535|98.666|27.781|
> |SPELL, shield radius=15|0.263|2.616|0.454|0.558|0.762|9.558|100.709|27.811|

---

> ### Author Response · Authors · 2024-11-14
> **Reviewer EnjK: Rebuttal discussion (part 2)**
>
> > **For instance, an analysis of average wall-clock time compared with baseline methods, or testing with larger reference dataset sizes, would be helpful for readers.**
>
> _**When using ImageNet-1k as the reference set (L.983 in our algorithm)**_, this is already detailed in the rightmost column of Table 2, L. 492. In that experiment with a reference set of over 1M images, using the same model, generation time goes from 3.5s to 6s using SPELL.
>
> The overhead of 2.5s is mostly due to the fast nearest-neighbor (NN) search to the $K=1,281,167$ reference set images at each $t$ during generation. That time grows _at most_ linearly with $K$ when adding more neighbors.
>
> We have extended **Table 2 with more parameters** (see revision), where we control more accurately the quality of fast-NN search by increasing the number of Voronoi cells that the NN algorithm calculates distances for, which controls both accuracy and compute. As shown below, when going from 1 to 2 cells, runtime increases by ~1.4 seconds, when going from 2 to 3 another 1.7s, from 3 to 5 by 2.2s and from 5 to 10 by 3.6 seconds. Note also that with more inference runtime, we are able to guarantee even better protection rates (0.16% at 10 cells compared to 0.6% in the paper, which used 2 cells). We have not optimized yet the parameters of the fast-NN search (currently this is run, sub-optimally, on CPU). We believe these overheads can be significantly reduced.
>
>
> | Model     	| Searched cells | Time per image (s) ↓ | Generated images too close to ImageNet neighbors ↓ |
> |---------------|----------------|-----------------------|-----------------------------------------------------|
> | EDMv2     	| -          	| 2.434             	| 7.60%                                          	|
> | + SPELL   	| 1          	| 4.633             	| 1.08%                                          	|
> | + SPELL   	| 2          	| 6.057             	| 0.55%                                          	|
> | + SPELL   	| 3          	| 7.790             	| 0.33%                                          	|
> | + SPELL   	| 5          	| 9.949             	| 0.22%                                          	|
> | + SPELL   	| 10         	| 13.545            	| 0.16%                                          	|
>
>
> _**When the reference set is a set of concurrently or previously generated samples (L.991 in our algorithm)**_ to increase diversity, the overhead consists of calculating pairwise distances for up to 128 (expected or realized) images at each diffusion time. That computation is negligible. The runtimes of the base model, SPELL, and also CADS, IG, and PG are equal in this setup.
>
> > **For example, the fourth image is repelled from the third image, and it's unclear why this image is closer to the third image than to the first or second.**
>
> This is likely due to background similarity between images 3 & 4 (top row). Note that SPELL intervenes *during generation* and not as a post-processing mechanism: To insist (in the spirit of L.425~), SPELL was *not* used because SD3 Image 4 came out as too close to Image 3; SPELL was used before that, during the generation of Image 4, because (L.463) it “was _expected_ to come out too close to the 3rd image”.
>
> > **Additionally, it would be beneficial if the authors provided examples with multiple image batches, other than a single-image batch.**
>
> Thanks for the suggestion! We plan to generate further examples (we are working to get access to compute resources) and will post these examples here very soon.
>
> > **I wonder whether the L2 distance-wise nearest neighbor search was the best choice. This is because many images in the third row (EDM + SPELL) seem more similar to the second row (ImageNet neighbor for EDM) rather than the fourth row (ImageNet neighbor for EDM+SPELL).**
>
> Your question can be parsed in 2 different ways: (i) whether to use the L2 distance, and (ii) whether to use L2 of images in the MDTv2 latent space to guide SPELL.
>
> On (i), Fast NN search (e.g. FAISS as used here) is designed to work for L2, and we are hence somewhat stuck with it.
>
> On (ii), indeed, we are not bound to using that representation for SPELL, the user may define any other encoding of interest. In image protection, defining an encoding also defines explicitly the type of protection that is achieved.
>
> For now, our goal in Fig. 7 was to convey clearly that SPELL avoids obvious copies (e.g. images 2 3 4 5 6 7 8) that using EDMv2 directly would output.
>
> > **However, according to Table 1, isn't this actually the case for 4 out of 6 models? Please correct me if I am wrong.**
>
> We apologize for not being clear. We write _The third diversity metric, coverage, is also increased in all models except SD3_ (L.323). This is technically correct, because SD3-Medium is considered both in the text2image and class2image tasks. But you are also correct, this is 4/6 _experiments_ in Table 1, we have corrected this entire part in L.358.

---

> ### Comment · Reviewer_EnjK · 2024-11-18
>
> I deeply appreciate the authors for the additional experiments, which have significantly clarified my understanding. That said, I still have a few unresolved questions:
>
> - Regarding Fig. 6: I understand that the 4th image is repelled from the 3rd due to the background. However, it seems that
>  the altered trajectory of the 4th image share the same background as the original (unaltered path) image. If the 4th image in (SD3+SPELL) had a different background, I would have no doubts about SPELL’s performance. Did I misunderstand something?
>
> - Thank you for the details on wall-clock inference time and the additional trade-off graph in Fig. 4. I now wonder if SPELL is competitive in inference speed compared to the baselines. This is because SPELL seems to be quite costly in Table 2. Could the authors provide another trade-off graph including wall-clock time? I understand this was not part of my initial questions, and I apologize for raising a new point.
>
> - I look forward to seeing the multi-batch results.

---

> > ### Author Response · Authors · 2024-11-18
> >
> > Thank you for joining the discussion! We appreciate your comments and are happy to enhance the revised paper following your suggestions.
> >
> > > **Regarding Fig. 6: It seems that the altered trajectory of the 4th image share the same background as the original (unaltered path) image. If the 4th image in (SD3+SPELL) had a different background, I would have no doubts about SPELL’s performance. Did I misunderstand something?**
> >
> > In this particular case, it does seem that the intervention was small. Any interpretation (as the one we made with background in our previous answer) is just an educated guess, as the geometric considerations that triggered SPELL are in latent space.
> >
> > However, an interesting point that you highlight is that for intra-batch diversity, or diversity to previously generated images, the most interesting and visible interventions will tend to happen **after** a few images have been generated, because there are more shields to take into account. In that sense, we believe that it is the **second** row in Fig. 6 that conveys best the results of SPELL interventions. This is very visible in iterations 11,14 or 15.
> >
> > This is what we wrote in the original caption and in the text, notably in **L.455**, _“Early images are adjusted less often and mostly in details because they are still novel enough. Later images repel from more previous images and more strongly to ensure they are different enough.“_, or in the discussion L. 458-464.
> >
> > > **Thank you for the details on wall-clock inference time and the additional trade-off graph in Fig. 4. I now wonder if SPELL is competitive in inference speed compared to the baselines. This is because SPELL seems to be quite costly in Table 2. Could the authors provide another trade-off graph including wall-clock time?**
> >
> > This is not a problem at all! We are happy to clarify first two important points:
> > The results for Table 2 (in **Section 4.6**) are for the **protection** task where the goal is to avoid recreating an image too close to the $K=1.2M$ images in imagenet (see. Fig. 7 for an illustration). The overheads are significant because of the sheer size of $K$, and entirely spent on a fast NN lookup. While we believe they could be brought down significantly with additional optimizations (search is currently done on CPU, which is not optimal), we believe the scale there is bound to cause some overhead.
> >
> > The experiments in Fig.4 (in **Section 4.3**) are for the **generation diversity** task, where the goal is avoid generating images self-similar to previously or concurrently generated images. The scale of the overhead in that case is **truly negligible** when contrasted with the diffusion generation cost (= computing up to ``[B,K]`` distance matrices per time $t$, where both ``B`` and ``K`` do not exceed hundreds, and adding one single correction vector to the score).
> >
> > As a result, unless a user wishes to explore self-diversity to a previously generated batch of 1M images, the timings in Table 2 have no relevance to the self-diversity experiments.
> >
> > **When looking at intra-batch diversity, we are happy to report the following timings, which agree with intuition presented above**. These timings represent generation time per second, using batches of size ``B=8``. As we expected, the overheads for SPELL and the other batch-diversity methods we benchmarked are negligible when using small batches. We have added this as Table 5 to Appendix G and refer to it in the main text in Section 4.6.
> >
> > |Model|Generation time per image (seconds)|
> > |------------------|----------------|
> > |Baseline (Simple Diffusion)|2.93±0.12|
> > |Simple Diffusion + PG|2.96±0.13|
> > |Simple Diffusion + IG|2.93±0.12|
> > |Simple Diffusion + CADS|2.96±0.12|
> > |Simple Diffusion + SPELL|2.94±0.13|
> >
> >
> > > **I look forward to seeing the multi-batch results.**
> >
> > Thanks for the suggestion and your patience. **We have now finished generating 400 further example images. We provide these examples of Simple Diffusion with and without SPELL in Figures 25 to Figure 34 in the revised paper.** The 10 prompts are chosen from MS COCO, which Simple Diffusion was not trained on. As opposed to Figure 1, this features both of SPELL’s capabilities: Intra-batch repellency (every row is a batch of 4), and inter-batch repellency from previous batches, which we treat as the shielded set. The examples affirm qualitatively that SPELL increases the diversity of generated images. Notably, this is without introducing visual artifacts and without lowering the prompt adherence, which other baselines like IG are prone to, see Table 4 and Figure 4.

---

> > > ### Comment · Reviewer_EnjK · 2024-11-19
> > >
> > > Thank you for your prompt response. Most of my concerns have been addressed. I thus raise my score from 5 to 6 and lean towards acceptance.

---

> > > > ### Author Response · Authors · 2024-11-19
> > > > **Many thanks for reading our rebuttal.**
> > > >
> > > > We are grateful for your grade increase, and we remain available to answer other concerns or requests for clarifications!
> > > >
> > > > Authors

---

### Author Response · Authors · 2024-11-22
**General response, a few days before rebuttal closing.**

Dear Reviewers,

We would like to thank you for your time. We felt very encouraged by your positive comments:
- our shielded generation addresses a _“timely and practically-relevant problem [...] with limited prior research, making this work particularly valuable”_ (EnjK, V3e4, 94st, H25b)
- our paper and notation is _“well-written and easy-to-understand”_ (H25b, EnjK, V3e4).
- our paper has “thorough empirical investigations” (94st, EnjK, H25b) that demonstrate that “SPELL achieves the best trade-off between image diversity and generation quality” (EnjK)
- our paper is among the first to enable image-level protection at large scale _“which is one of the significant challenges for real users”_ (94st, H25b, EnjK, V3e4).

More importantly, we are grateful for your many insights and questions. They have triggered a few minor modifications. We have updated our draft with all experiments and clarifications you have requested. They are highlighted in blue in the revised paper and include:
- An additional ablation that shows that with more inference-time runtime, SPELL shields all 1M+ protected images in 99.84% of the generated images
- 400 new example images on unseen prompts from MS COCO to qualitatively study SPELL’s increased diversity (Figures 25-34)
- Extended runtime results in both the diversity (Table 5) and the protection experiment (Table 2) that show that SPELL’s cost is negligible in small batches and scales sub-linearly when shielding 1M+ images simultaneously
- A fourth trade-off plot that investigates the FD_DINOv2 in Figure 4
- Table 4 which shows all 8 metrics that we have tested in the trade-off experiment for all hyperparameter combinations of SPELL and the three recent baselines
- A more detailed comparison between the intra-batch component of SPELL and  Particle Guidance,
- Standard deviations across five seeds for all six diffusion models in Table 1

We are very grateful for Reviewer **EnjK**'s interaction, and for their subsequent decision to increase their general score $5\rightarrow 6$.

We remain available for further requests for clarification or questions, for the remainder of the rebuttal period.

The Authors

---

### Meta-Review · Area_Chair_JGeL · 2024-12-23

**Metareview:**

This paper proposes a technique for shielding the diffusion model from protected images by adding sparse repellency terms to the diffusion process. SPELL is a post-training intervention that enhances image diversity and safeguards specific images by selectively adjusting generation paths. The problem of shiedling generated images from a set of protected images is a very important problem that needs to be studied. This paper shows some quantitative and qualitative results demonstrating the appraoch.

**Additional Comments On Reviewer Discussion:**

One of the main concerns that reviewers (and I myself) have is the lack of sufficient experimental results, trade-offs, quantitative analysis, challenging text to image cases, etc. The authors addressed some of these concerns in the rebuttal which I appreciate. But, I feel the authors could have done much better in showing the effectiveness of the approach.

Instead of just using Imagenet and COCO where it is hard to see the effect of guardrailing, the authors could have use face datasets and shown if some identities are not generated. Use of face recognition networks can give a concrete score for this.
An analysis on size of shielding dataset would be nice to have. For example, if I only have one image per face subject, is it sufficient? Do I need to have multiple samples per identitity for the approach to work well, etc?
Some examples on SOTA high resolution image generators like Flux on challenging use cases would be nice.

I feel like the paper as such is borderline (excluding reviewer V3e4). Including more concerete qualitative examples would make the paper very strong. So, as such, I regret to inform that the paper is not in a state to accept to a competitive conference like ICLR. I would really encourage authors to take into account all the feedback and resubmit with a strong submission.

---

### Decision · Program_Chairs · 2025-01-22

Reject